# Protein engineering and iterative multimodule optimization for vitamin B$_6$ production in *Escherichia coli*

Linxia Liu [1,2,3,6], Jinlong Li [1,3,4,6], Yuanming Gai[1,2,6], Zhizhong Tian[1], Yanyan Wang[1], Tenghe Wang[1], Pi Liu [1], Qianqian Yuan [1], Hongwu Ma [1], Sang Yup Lee [5] ✉ & Dawei Zhang [1,2,3,4] ✉

Vitamin B$_6$ is an essential nutrient with extensive applications in the medicine, food, animal feed, and cosmetics industries. Pyridoxine (PN), the most common commercial form of vitamin B$_6$, is currently chemically synthesized using expensive and toxic chemicals. However, the low catalytic efficiencies of natural enzymes and the tight regulation of the metabolic pathway have hindered PN production by the microbial fermentation process. Here, we report an engineered *Escherichia coli* strain for PN production. Parallel pathway engineering is performed to decouple PN production and cell growth. Further, protein engineering is rationally designed including the inefficient enzymes PdxA, PdxJ, and the initial enzymes Epd and Dxs. By the iterative multimodule optimization strategy, the final strain produces 1.4 g/L of PN with productivity of 29.16 mg/L/h by fed-batch fermentation. The strategies reported here will be useful for developing microbial strains for the production of vitamins and other bioproducts having inherently low metabolic fluxes.

Vitamin B$_6$ comprises of six interconvertible pyridine compounds, including pyridoxine (PN), pyridoxamine (PM), pyridoxal (PL), and their phosphorylated derivatives, pyridoxine 5′-phosphate (PNP), pyridoxamine 5′-phosphate (PMP), and pyridoxal 5′-phosphate (PLP)[1–3]. PLP is an essential cofactor of enzymes required by >160 cellular reactions including transamination, decarboxylation, β-elimination, β-substitution, γ-elimination, and γ-substitution[4–6]. These reactions are involved in diverse pathways including amino acid and neurotransmitter metabolism, folate and 1-carbon metabolism, sugar and lipid metabolism, and erythropoiesis[7,8]. Because of the vital role of PLP in these pathways, PLP deficiency causes severe effects, including seborrheic dermatitis, convulsions, microcytic anemia, peripheral neuropathy, epilepsy, and depression[7,9,10]. Plants and microorganisms have a naturally de novo biosynthetic pathway for vitamin B$_6$, but humans and animals have to be externally obtained from dietary supplements or feed additives[11]. Thus, vitamin B$_6$ supplementary diet is of high commercial interest in pharmaceutical, food industries, and livestock feed[12,13]. The intestine only absorbs non-phosphorylated B$_6$ vitamers, which are converted by specific enzymes to the active PLP form[7]. PN hydrochloride is the most common commercial form of vitamin B$_6$, which is currently produced through complete chemical synthesis[12]. The chemical process used too many solvents such as benzene and toluene, which belong to Class 1 and 2 solvents labeled in the United States Pharmacopeia (USP), respectively. And, the industrial by-products were difficult to treat and costly, such as large amounts of phosphate. Additionally, high-temperature and chlorination reactions increased the safety risk[9]. Furthermore, national environmental protection supervision and control have become essential means to stimulate green technology innovation[14]. Given these factors, there is a

[1]Tianjin Institute of Industrial Biotechnology, Chinese Academy of Sciences, Tianjin, China. [2]National Technology Innovation Center of Synthetic Biology, Tianjin, China. [3]Key Laboratory of Engineering Biology for Low-Carbon Manufacturing, Tianjin Institute of Industrial Biotechnology, Chinese Academy of Sciences, Tianjin, China. [4]University of Chinese Academy of Sciences, Beijing, China. [5]Department of Chemical and Biomolecular Engineering (BK21 four program), Korea Advanced Institute of Science and Technology (KAIST), 291 Daehak-ro, Yuseong-gu, Daejeon 34141, Republic of Korea. [6]These authors contributed equally: Linxia Liu, Jinlong Li, Yuanming Gai. ✉e-mail: leesy@kaist.ac.kr; zhang_dw@tib.cas.cn

pressing need for an alternative fermentation-based production method for vitamin $B_6$.

Vitamin $B_6$ is biosynthesized de novo by two different pathways, the deoxyxylulose 5-phosphate (DXP)-dependent pathway and the DXP-independent pathway (Fig. 1)[2]. The longer DXP-dependent route consists of two branches and seven enzymes, which is mainly present in α- and γ-proteobacteria, such as the Gram-negative model bacterium *Escherichia coli*[13,15]. In this pathway of *E. coli*, PNP phosphatase has not been identified and may use phosphatase with a broad substrate spectrum[16]. The shorter DXP-independent biosynthetic pathway only involves the PdxST enzyme complex and is found in most eubacteria, including the Gram-positive model bacterium *Bacillus subtilis*, fungi, protozoa, archaea, plants, and metazoans[11,17,18]. To date, no organism

with both vitamin $B_6$ biosynthetic pathways has been identified, but many organisms including mammals and humans harbor a salvage pathway for the interconversion of $B_6$ vitamers (Fig. 1)[19].

Attempts to biosynthesize vitamin $B_6$ have been ongoing for more than 50 years[20]. Initially, *Ensifer meliloti* (formerly *Sinorhizobium meliloti* or *Rhizobium meliloti*) IFO14782 was found to naturally produce ~100 mg/L of vitamin $B_6$ in 168 h of fermentation. PN production could be further increased to 1.3 g/L by overexpressing its *pdxJ* gene (encoding PNP synthase) and *E. coli epd* gene (encoding E4P dehydrogenase)[21,22]. However, due to the slow growth of *Ensifer*, the productivity was only 7.74 mg/L/h. In addition, the lack of effective gene manipulation techniques limited further research progress of *Ensifer* as a chassis cell to construct an efficient vitamin $B_6$ cell factory.

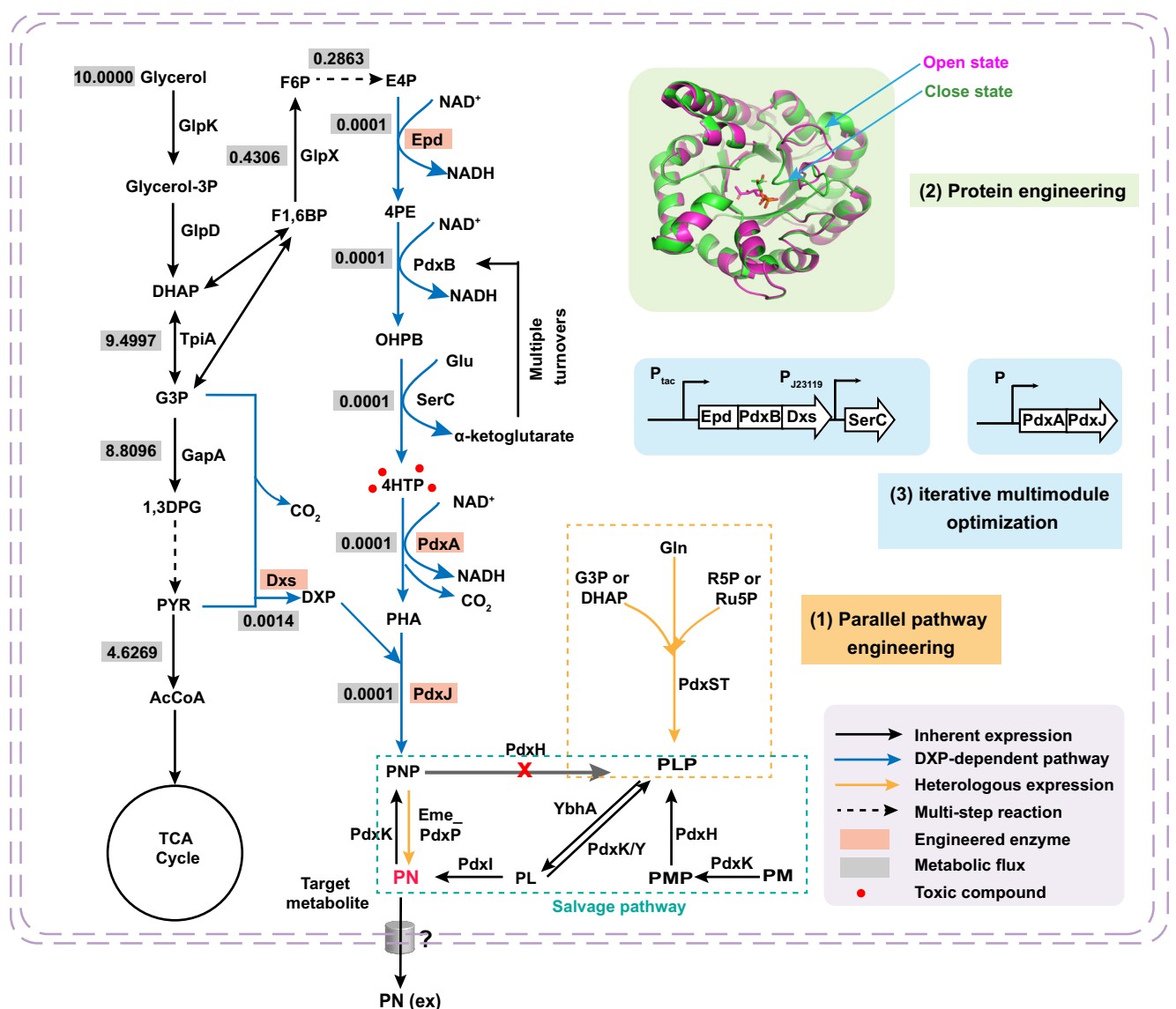

**Fig. 1 | Major metabolic pathways associated with PN biosynthesis in *E. coli*, metabolic engineering approaches applied to overproduce PN, and carbon flux from glycerol toward PNP starting from glycerol.** Blue arrows indicate the DXP-dependent PN biosynthetic routes in *E. coli*. The orange dotted box represents a heterologous DXP-independent pathway. The deleted gene is labeled with a red cross. Gray squares represent the metabolic fluxes for maximum biomass formation, and the numbers in squares represent the metabolic fluxes of reactions. The uptake rate of glycerol was defined as 10 mmol gDCW⁻¹ h⁻¹. Source data are provided as a Source Data file. Enzymes: Epd erythrose 4-phosphate dehydrogenase, PdxB 4-phosphoerythronate dehydrogenase, SerC 3-phosphoserine

aminotransferase, PdxA 4-phosphohydroxy-ʟ-threonine dehydrogenase, PdxJ PNP synthase, Dxs 1-deoxyxylulose 5-phosphate synthase, PdxP PNP phosphatase, PdxH PNP oxidase, PdxK PN/PL/PM kinase, PdxY PL kinase, PdxI PL reductase. Metabolites: E4P erythrose 4-phosphate, 4PE 4-phosphoerythronate, OHPB 2-oxo-3-hydroxy-4-phosphobutanoate. 4HTP 4-phosphohydroxy-ʟ-threonine, PHA 3-phosphohydroxy-1-aminoacetone, DHAP dihydroxyacetone phosphate, G3P glyceraldehyde 3-phosphate, 1,3DPG 1,3-bisphospho-ᴅ-glycerate, Pyr pyruvate, AcCoA acetyl-CoA, Glu glutamate, Gln glutamine, R5P ribose 5-phosphate, Ru5P ᴅ-ribulose 5-phosphate. Species: Eme *Ensifer meliloti*.

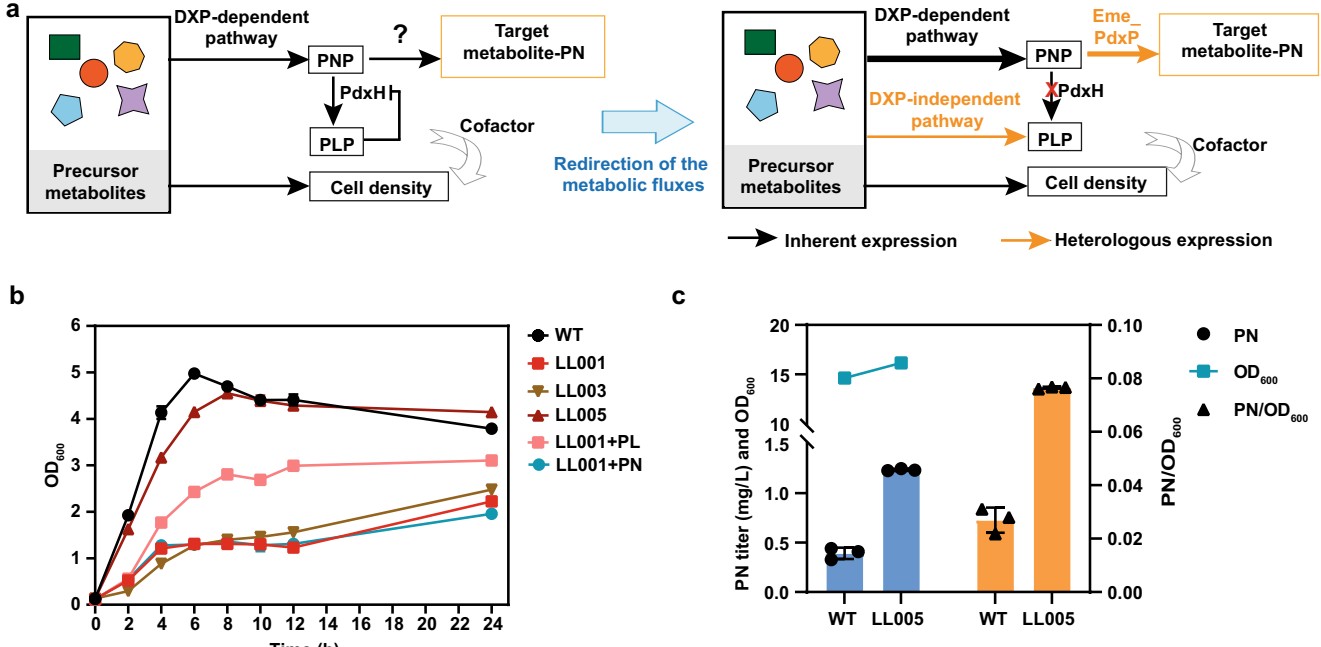

**Fig. 2 | Parallel metabolic pathway engineering of vitamin B6 de novo biosynthetic pathways. a** Schematic illustration of the strategy for parallel pathway engineering. The thickness of the arrows represents the proportion of carbon flow. Red crosses indicate the disruption of metabolic pathways. **b** The cell growth of the $\Delta pdxH$ mutants (LL001) with or without $pdxST$ expression. LL003, $\Delta pdxH::pdxST$-1; LL005, $\Delta pdxH::pdxST$-2. PN or PL was added to the medium at a concentration of 1 μM. **c** The PN titer, $OD_{600}$, and the values of $PN/OD_{600}$ of wild type (WT) and LL005. **b, c** Error bars here represent the standard deviation of three biological replicates ($n = 3$). Source data are provided as a Source Data file.

In a different study, an engineered *E. coli* strain overexpressing the native *epd*, *pdxJ* and *dxs* (encoding DXP synthase) genes produced 79 mg/L vitamin B6 (a mixture of vitamers) in 31 h[23]. Also, *B. subtilis* equipped with an exogenous DXP-dependent pathway produced 14 mg/L PN in a small-scale production assay[24]. *B. subtilis* was also engineered to convert the toxic metabolite 4-hydroxythreonine (4HT) to PN by expressing the *pdxA* (encoding 4HTP dehydrogenase) and *pdxJ* genes from *E. coli*. The resulting strain produced 65 mg/L PN by cofeeding 4HT and an amino acid cocktail (Asp, Lys, Thr, Met, and Ile)[25]. Unfortunately, none of these showed high enough efficiency for the fermentative production of vitamin B6[26].

Since the PdxST (PLP synthase PdxS and glutamine aminotransferase PdxT) enzyme complex of the DXP-independent pathway is rather inefficient and difficult to improve by rational design[13], the DXP-dependent pathway is more suitable for the production of vitamin B6. By comparing the kinetic properties and the expression level of the enzymes of the *E. coli* DXP-dependent pathway, it was revealed that vitamin B6 is produced by a series of "sluggish" enzymes[1,13]. The key enzymes including Epd, PdxA, and PdxJ showed low turnover numbers and high $K_m$ values and were present at low levels. Furthermore, in silico genome-scale metabolic simulation using iML1515[27] suggested low metabolic flux through the DXP-dependent pathway (Fig. 1). These multiple problems make it challenging to enhance the titer and productivity of vitamin B6[1,27–29]. Furthermore, engineered cells require delicate metabolic regulation to avoid the accumulation of the toxic intermediate 4HTP, which impairs growth[25,30]. Therefore, to develop an efficient PN-producing strain, it is necessary to address the challenges described above.

In this work, we establish an *E. coli* chassis for an efficient de novo biosynthesis of PN. Parallel metabolic pathway engineering of vitamin B6 biosynthetic pathways is performed to decouple cell growth-required PLP synthesis and PN production. Then, sequence- and structure-guided protein engineering is carried out to improve the key enzymes of the vitamin B6 biosynthetic pathways, including the starting enzyme Epd/Dxs and the inefficient enzymes PdxA/PdxJ. Subsequently, iterative multimodule optimization is performed to combine genes and balance the expression of up- and down-modules to maximize PN production. The final engineered *E. coli* strain produces 1.4 g/L of PN by fed-batch fermentation in a 5 L bioreactor.

## Results

### Parallel metabolic pathway engineering of vitamin B6 de novo biosynthesis pathways

Efficient pathway engineering for vitamin B6 biosynthesis is very important to release the tight regulation in the engineered strain. As the cofactor of a great number of enzymes involved in central metabolic pathways, PLP is the catalytically active form of vitamin B6, but it is toxic at high concentrations due to its high reactivity[31,32]. It is directly biosynthesized by the DXP-independent pathway, while PNP is the first B6 vitamer formed by the DXP-dependent pathway (Fig. 1). PNP oxidase PdxH, which catalyzes the oxidation of PNP to form PLP, plays a key role in controlling the intracellular homeostasis and bioavailability of PLP[33,34]. It has been demonstrated that PdxH is subject to mixed-type feedback inhibition by PLP via direct binding at an allosteric site[33]. PLP is needed only in a very small catalytic amount, and some enzymes store PLP, which can be transferred to PLP-dependent apoenzymes[1]. Coupled with inefficient enzymes, the metabolic flux through the vitamin B6 biosynthesis pathway is very low (Fig. 1). Considering the direct products of biosynthetic pathways, a growth decoupling strategy was used to rapidly accumulate biomass without inhibiting product formation[35]. Parallel metabolic pathways were engineered in which one pathway produces the target commercial chemical PN, while another pathway supplies PLP essential for cell growth and maintenance (Fig. 2a). The two pathways are separated by the deletion of PNP oxidase to block the transformation of PNP into PLP, allowing the two pathways to be metabolically orthogonal in the same strain without interfering with each other and cell growth.

In this study, different *E. coli* strains were evaluated for PN production, including MG1655, BW25113, JM109 (DE3), DH10B, Turbo, and AD494 (DE3). Glycerol, rather than glucose, was used as the main carbon source in the fermentation since the metabolic pathway is shorter to G3P, the direct precursor of PN[36]. The results showed that the production of PN was between 0.03 and 0.05 mg/L (Supplementary Fig. 1a). *E. coli* MG1655, with better performance, was used as the starting strain for the biosynthesis of vitamin B₆.

To decouple PN production from cell growth, we first knocked out the native *pdxH* gene encoding PNP oxidase to obtain strain LL001. Elimination of the *pdxH* gene resulted in poor growth due to a lack of PLP and accumulation of PNP (Supplementary Fig. 1b), which disrupts the glycine cleavage (GCV) system and caused the synthetic lethality[37]. The mutant was partly but not fully restored by supplying 1 µM PL, which could not be satisfied by 1 µM PN (Fig. 2b). For the native supply of PLP, the *pdxST* genes encoding enzymes from *B. subtilis* that directly synthesize PLP were inserted into the *pdxH* locus using the CRISPR-Cas9 system[38]. The *pdxST* genes were expressed in a single operon, under the control of the constitutive Biobrick J23118 promoter, which is considered to impose the lowest growth burden in the family of constitutive promoter parts (http://parts.igem.org/Part:BBa_J23118). Furthermore, the ribosome binding site (RBS) strength can be modulated to fine-tune gene expression levels, resulting in an engineered strain that grew similarly to the WT. The RBS-controlled translation strength of PdxST expression was optimized by the RBS calculator using moderate and weak strength (Supplementary Table 1). We obtained two engineered strains (named LL003 and LL005). When all the strains were cultured in an LB medium (as shown in Fig. 2b), LL005 showed a growth rate similar to that of the WT strain. The results showed that overexpression of *pdxST* from *B. subtilis* compensated for the growth defect of *ΔpdxH* by providing it with PLP necessary for growth (Fig. 2b). The final strain LL005 formed a parallel metabolic pathway for vitamin B₆ de novo biosynthesis pathways based on the direct synthetic B₆ vitamer and was found to be enhanced PN production through the redirection of metabolic flux from PNP towards PN, rather than to PLP (Fig. 2c and Supplementary Fig. 1b).

Additionally, PNP phosphatase catalyzes the last step of PN biosynthesis in *E. coli*, which is unknown (Fig. 1). As reported previously, *pdxP* from *E. meliloti* encodes a relatively specific PNP phosphatase[39,40]. To complete the PN formation pathway and avoid the accumulation of PNP which can perturb amino acid metabolism[8], the *pdxP* gene was inserted into the *pta* locus of *E. coli* in the above-described strain LL005 with a parallel metabolic pathway. Pta (encoding phosphate acetyltransferase) disruption is no effect on the fermentation process for PN (Supplementary Fig. 1c), which also can reduce the production of undesirable acetate[41,42]. The obtained strain LL006 was used as a test strain for further enzyme design and optimization to increase PN production.

## Protein engineering of key enzymes

The efficiency of biosynthetic enzymes directly determines the rates of metabolic reactions and thus controls the flux distribution, which is linked to the yield of the final products[43,44]. However, while improving enzyme activity can greatly contribute to production yields, it may not guarantee a continuous increase in those yields. In this study, to maximize the yield, an enzyme design method[45] was applied to enrich more beneficial mutations in the construction of the mutation library. This library may encompass different mutations, each with high activity, which can be further screened to identify combinations of mutations that are well-suited to the pathway. Furthermore, for different enzymes, structure modification and sequence screening strategies were carried out, according to the availability of references (e.g., Dxs), structural information (e.g., Epd), and the accessibility of catalytic models (e.g., PdxA and PdxJ). Due to the limited availability of biosynthetic intermediates and better response to the impact of protein engineering on yield, the effect of specific mutations in a gene or gene mining was examined by indirectly measuring fermentation yield in vivo.

## Engineering the rate-limiting enzymes to increase the pull driving force

PdxA and PdxJ are the key enzymes in the DXP-dependent pathway (Fig. 1)[2]. PdxA converts the toxic intermediate 4HTP to PHA, which is highly unstable[46,47]. Therefore, enzymatic analysis of PdxJ usually has to be coupled to that of PdxA, resulting in the highly complex and challenging derivation of kinetic parameters that are complicated[2]. PdxJ is the last enzyme in the de novo biosynthesis of vitamin B₆, catalyzing the complicated ring-closure reaction between DXP and PHA (Fig. 1)[32]. It was reported that PdxJ is the rate-limiting enzyme of the vitamin B₆ biosynthesis pathway[26]. The kinetic parameters for PdxJ were determined indirectly by coupling the reaction with PdxA, which yielded a $K_m$ for DXP of 26.9 µM and a $k_{cat}$ of 0.07 s⁻¹, which is a quite low turnover number[48]. To increase the pull driving force for PN production, a structure-based computational enzyme redesign method was applied to improve the performance of PdxA and PdxJ.

PdxA forms tightly bound dimers with the active site located at the dimer interface, within a cleft between the two subdomains and involving residues from both monomers[46]. To construct the catalytic model, 4HTP and NAD⁺ were docked into the binding pocket according to the crystal structure PDB 1PS6 and PDB 6XMY[46]. The catalytic mechanism was based on malate dehydrogenase (oxaloacetate-decarboxylating)[49,50]. A complex structure with the protonation-converted substrate (2-amino-3-oxo-4-phosphonooxybutyrate, APB) was modeled to mimic the transitional state for enzyme design[45]. Finally, the residues within 8 Å around the intermediate were obtained for the simulation of iterative saturation mutagenesis, and the mutants that have a better binding energy than the WT, as well as those with the best binding energy in each position, were selected for experimental verification. The *pdxA* gene was first codon-optimized to remove its rare codons and then inserted into the pRSFDuet-1 expression vector. The mutations with better binding energy were introduced into the codon-optimized *pdxA* coding sequence, and the resulting plasmids were introduced into the LL006 chassis strain. Cultivation in deep-well plates was used for the screening of PdxA mutants with enhanced PN production as measured by HPLC. To improve the yield of PN, two rounds of enzyme redesign were performed on PdxA. The improved mutants (H136N, P245C, V149L, and T165R/T285L in Fig. 3a) showed a ~100% increase in PN production compared with the WT *pdxA* overexpression strain, with titers exceeding 2.0 mg/L (all 128 mutants are listed in Supplementary Data 1).

To elucidate the possible effects of these beneficial mutations on PdxA, the Rosetta-designed structure was used for further analysis. The results showed that only H136N interacted directly with the substrate. Other beneficial mutations did not act directly on the substrate but formed new interactions locally (Supplementary Fig. 2), which may indirectly stabilize the structural domain bound to the substrate. These factors may favor the stable catalysis of PdxA and thus enhance the yield. In addition, although mutant residue N136 interacted with the substrate, WT H136 originally formed hydrogen bonds with the substrate. Therefore, the possible binding differences between H136N and WT were analyzed by MD simulations. The binding energies of H136N and WT were −174.00 and −163.16 kcal/mol, respectively. The computed binding energy of H136N was approximately 10.84 kcal/mol smaller than that of the WT. Further decomposition of the binding energy revealed that the binding energies of residue 136 in H136N and WT were −27.08 and −20.11 kcal/mol, respectively. Thus, the change in the binding energy of 6.97 kcal/mol implies that most of the overall binding energy change is attributed to the contribution of residue 136. Representative conformational analysis revealed that the H136N mutation may cause a change in the substrate−protein interaction

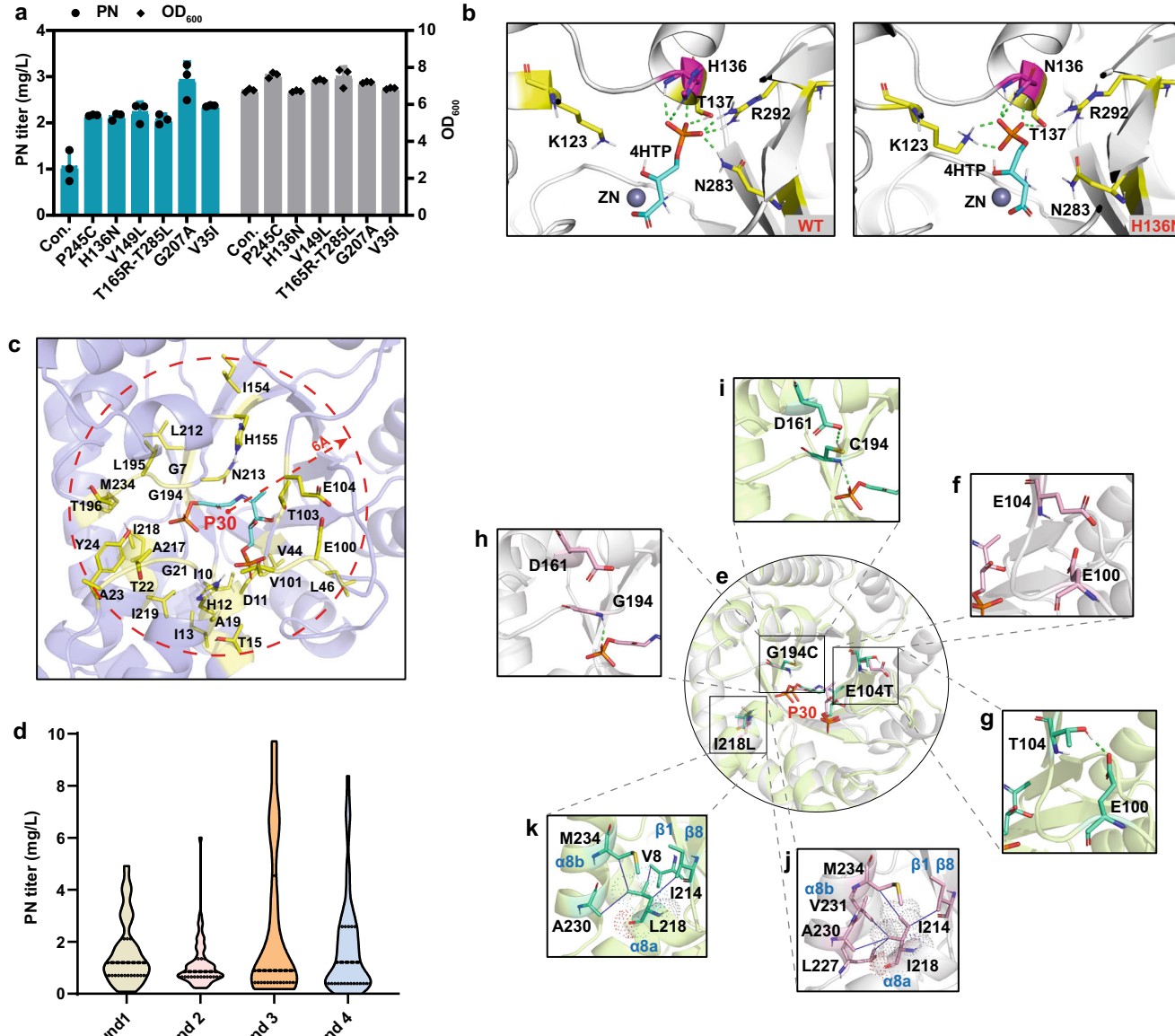

**Fig. 3 | Rational design of inefficient enzymes PdxA and PdxJ. a** The PN titer and cell growth (OD$_{600}$) of WT enzyme (Con.) and *pdxA* mutants. Data are presented as mean values ± SD from three independent biological replicates (*n* = 3), the circles or diamonds represent individual data points. **b** Comparison of the binding conformations of 4HTP in WT and H136N based on molecular dynamics (MD) analysis. The dotted green line indicates hydrogen bonding interactions. **c** Residues (yellow sticks) within a 6 Å radius from the imine intermediate P30 (cyan sticks). **d** The PN titer distribution of four rounds of PdxJ mutants including a total of 212 mutants:

26 single mutants (round 1), 62 double mutants (round 2), 60 triple mutants (round 3), and 64 quadruple mutants (round 4). All the detailed data are listed in Supplementary Data 3. **e** Distribution of the mutation sites (E104T/G194C/I218L) in LL239. **f** and **g**, **h** and **i, j** and **k** Differences before and after introducing the E104T, G194C, and I218L mutations, respectively. Substrates and residues are represented as pink in the WT and green in mutant structures. The dotted green line indicates hydrogen bonding interaction and the solid blue line indicates hydrophobic interaction. Source data are provided as a Source Data file.

network (Fig. 3b). Furthermore, the H136N mutant did not increase the protein expression (Supplementary Fig. 3a), which suggested that the increase in production yields may be due to the increase in enzyme catalytic efficiency of H136N.

To further improve the PdxA properties, we found that the PdxA protein from *E. coli* exhibits poor hydrophobic packing in its predicted structure. The hydrophobic effect plays a central role in protein stability, and defects in hydrophobic core packing are associated with reduced stability[51]. We augmented the packed hydrophobic cores to favor protein stability using in-house scripts (https://github.com/Gerald-li/Protein_Stability_design-workflow). A collection of 45 small, buried hydrophobic residues were replaced with larger ones. Among them, 17 amino acid mutations showed better stacking

of hydrophobic interactions than the WT control (Supplementary Data 2). Next, we superimposed the improved sites by region and produced 17 mutants, among which 10 engineered strains exhibited an improved PN yield (Supplementary Data 2). Overall, the individual mutations G207A and V35I had the best effect, with 166% and 119% increases in PN production, respectively (Fig. 3a). These two mutant side chains fill the unsaturated cavities in their vicinity better than the WT residues, resulting in superior hydrophobic interactions (Supplementary Fig. 4). Interestingly, the expression of G207A and V35I mutants was not increased (Supplementary Fig. 3a). It was possible that the mutants stabilized the local region of PdxA and then increased the catalytic efficiency, which led to an eventual increase in yield.

PdxJ is the bottleneck enzyme in the DXP-dependent pathway. The crystal structure of PdxJ (PDB 1M5W) showed that the octameric enzyme possesses eight distinct binding sites and three different binding states[52]. The correlation of the enzyme structure to binding site occupancy is mediated through the conformation of the active site loop composed of amino acids 95–105[52]. PdxJ provides apparently three structurally and functionally distinct states: (1) a resting, open state, (2) a partially open state with single-bound DXP, and (3) a closed state with bound DXP and inorganic phosphate, a surrogate of AHP positioned to react (Supplementary Fig. 5)[52]. Yeh et al. indicated that the closed state favored catalysis. Due to the many reaction steps of PdxJ, to mimic the transition state structure and retain enough binding space, the imine intermediate (P30, formed after the first step) was selected and docked into the binding pocket for the design module (Fig. 3c)[32]. Residues approximately 6 Å from P30 (conserved residues were excluded) were selected for enzyme design (Fig. 3c).

By optimizing the protein structure of the WT, the maximum binding affinity (binding energy = −7.95 kcal/mol) was taken as a reference for subsequent evaluation and selection of variants. The mutants with lower binding energy scores than the WT and the one with the best binding energy at the same residue were selected as the hits, which resulted in 26 mutants that were selected for experimental validation (G7L, I10L, D11C, H12M, I13V, T15I, V44I, E100I, V101L, T103L, E104T, I154L, H155I, G194C, L195I, T196C, L212I, N213I, A217V, I218L, I219G, A19Q, G21M, T22V, A23T, and Y24I). The mutated *pdxJ* gene was inserted into the expression vector pRSFDuet-1, and the resulting plasmids were introduced into LL006 chassis cells. The PN titer was determined by the HPLC method. Among these mutants, 10 showed a more than 2-fold titer increase compared with the original gene (Supplementary Fig. 6). As described in the section "Methods" under subsection "Computational enzyme redesign", four rounds of computational designs were simulated by using the positive mutants from the previous round of experimental results as a template. A total of 212 mutants were constructed, including 26 single mutations, 62 double mutations, 60 triple mutations, and 64 quadruple mutations. As shown in Fig. 3d, the titer of the quadruple mutation was not higher than that of the triple mutation mutant, so we terminated further mutation experiments (Fig. 3d and Supplementary Data 3). The yield of the strain expressing mutant E104T/I218L/G194C (LL239) was 18-fold higher than that of the control strain (Supplementary Data 3), and this mutant protein was named PdxJ1.

The structure of PdxJ1(E104T/I218L/G194C) was extracted from the Rosetta enzyme design results to investigate the possible effects. As shown in Fig. 3e, the side chains of the three residues do not have direct interactions with the substrate but are not distant. Residue 104 was located on the loop, which is related to the opening and closing of the protein and the stability of this loop is very important for catalytic activity. E104T formed a new hydrogen bonding interaction with E100, thus stabilizing the loop structure and helping to maintain the closed state (Fig. 3f and g). The G194C mutation can stabilize the loop structure by forming a new hydrogen bonding network with D161, which may stabilize substrate binding (Fig. 3h and i). Unlike the other two positions, I218 is in a hydrophobic environment. Although the mutation of I218L does not increase the number of hydrophobic interactions, the new mutation L218 increases the hydrophobic interaction with residue V8 from the β1 structure, making α8a and β1, β8 and α8b domain may form a more stable hydrophobic effect, while most hydrophobic interactions of I218 comes only from α8b domain (Fig. 3j and k). This may favor protein stabilization and increase expression. In addition, the structures of PdxJ1 and WT were simulated by MD, which revealed binding energies of −121.58 and −99.59 kcal/mol, respectively. It is noted that the computed binding energy of PdxJ1 was approximately 22 kcal/mol smaller than that of the WT. Additional research was carried out to investigate the protein expression levels in the mutants with higher yields, but no significant increase was observed. Conversely, the expression levels decreased significantly in some cases (E104T/I218L/G194C and I13V/L212I/E104T) (Supplementary Fig. 3b). These observations indirectly suggest that the enhanced production yields may be attributed to an increase in enzyme catalytic efficiency.

## Increasing the push of precursors toward the vitamin B$_6$ pathway

This first step of the vitamin B$_6$ biosynthesis pathway is believed to be rate-limiting in some organisms and represents a branch point in bacterial metabolism[53]. As such, 'pushing' the flux toward the biosynthetic pathway is important to enhance the overall yield[54–56]. The DXP-dependent vitamin B$_6$ biosynthesis pathway consists of two branches, whereby the first enzyme of the longer branch of this pathway is Epd, and the shorter branch is Dxs (Fig. 1). Candidate homologs of the *epd* and *dxs* genes identified by in silico mining could be a complementary strategy to broaden the branch pathways fluxes for vitamin B$_6$ production.

Epd, which catalyzes the NAD$^+$-dependent oxidation of E4P, is homologous to a key enzyme of the glycolytic pathway, with 42% amino acid sequence identity to G3P dehydrogenase (GapA) from *E. coli* (Eco_Epd)[57,58]. However, Epd shows efficient non-phosphorylating E4P dehydrogenase activity and low phosphorylating G3P dehydrogenase activity, while GapA shows the opposite catalytic specificity[1]. The apparent $K_m$ values of Eco_Epd reported in the literature are 0.51–0.96 mM for E4P and 0.074–0.8 mM for NAD$^+$[59,60]. The high $K_m$ means that the Epd enzyme has a low affinity for its substrate or cofactor and requires a greater concentration of the substrate to reach Vmax[61]. Given the different specificities and poor enzyme kinetics, genome-wide mining of *epd* homologous genes was performed to enhance PN production.

Mining of *epd* homologous genes was conducted based on sequence similarity coupled with crystal structural information. According to the annotation of Epd in the Swiss-Prot database and common strains, 251 sequences from different species were based on 3123 Epd sequences in the BRENDA database (EC 1.2.1.72)[62]. MEGA11 software[63] was used to create an evolutionary tree to analyze the differences between sequences. First, 20 *epd* genes from different sources were selected based on estimating the evolutionary divergence between sequences, using the selection criterion that the evolutionary difference between the selected sequences and the *E. coli* input sequence should be 12.6–68.4% (Fig. 4a). Protein structures were predicted using AlphaFold2, and the stability of these proteins was analyzed via the CNA web server[64]. Finally, 11 heterologous *epd* genes encoding enzymes that may have better structural stability were selected for gene synthesis.

The codon-optimized sequences were artificially synthesized and expressed in PdxJ-overexpressing *E. coli* (LL239) using the p15ASI vector backbone[65]. The PN titer on the strains expressing 11 heterologous *epd* genes expression was compared with the strain expressing the native *epd*. The results showed that all 11 selected genes exhibited diverse effects on PN biosynthesis in our expression systems, whereby the gene from *Glaciecola nitratireducens* resulted in a 2.2-fold increase of the product titer, reaching 29 mg/L (Fig. 4b). To elucidate possible reasons for the high yield of Gni_Epd, the structures of Gni_Epd and Eco_Epd were simulated by MD. The binding energies of NAD$^+$ in Gni_Epd and Eco_Epd were −64.04 and −53.34 kcal/mol, respectively. It is noted that the computed binding energy of NAD$^+$ in Gni_Epd was -11 kcal/mol smaller than that of NAD$^+$ in Eco_Epd (Supplementary Table 2). The difference in binding energy between Gni_Epd and Eco_Epd may be due to the great change in the binding mode of NAD$^+$, especially the nicotinamide moiety of NAD$^+$ and the sugar ring attached to it (Fig. 4c). We also found that the binding modes of E4P in Gni_Epd and Eco_Epd are different, and the binding energy of E4P (−13.11 kcal/mol) was

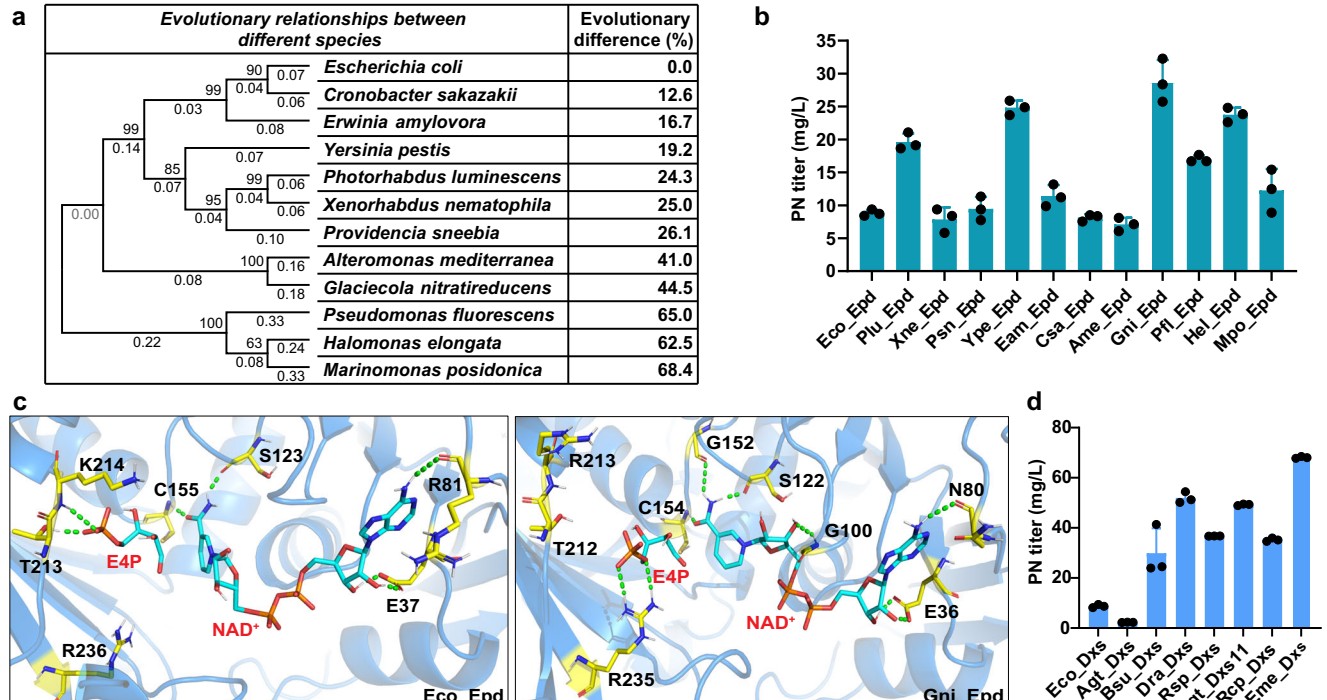

**Fig. 4 | Mining of *epd* and *dxs* genes to enhance PN production. a** The evolutionary difference between *epd* genes and the *E. coli* input sequence. **b** The PN titer of *epd* expression in the LL239 strain. Eco *E. coli*, Plu *Photorhabdus luminescens*, Xne *Xenorhabdus nematophila*, Psn *Providencia sneebia*, Ype *Yersinia pestis*, Eam *Erwinia amylovora*, Csa *Cronobacter sakazakii*, Ame *Alteromonas mediterranea*, Gni *Glaciecola nitratireducens*, Pfl *Pseudomonas fluorescens*, Hel *Halomonas elongata*, Mpo *Marinomonas posidonica*. **c** The binding modes of E4P and NAD⁺ in Eco_Epd (left panel) and Gni_Epd (right panel), respectively. **d** The PN titer of *dxs* expression in the LL239 strain. Agt *Agrobacterium tumefaciens*, Bsu, *B. subtilis*, Dra *Deinococcus radiodurans*, Rsp *Rhodobacter sphaeroides*, Rcp *Rhodobacter capsulatus*, Eme *E. meliloti*. **b, d** Data are presented as mean values ± SD from three independent biological replicates (*n* = 3), the circles represent individual data points. Source data are provided as a Source Data file.

slightly higher than that of Eco_Epd (−12.54 kcal/mol), suggesting some stabilization binding of E4P. Consistent with the calculated binding energy results, the enzyme kinetic data indicated that the binding capacities of both E4P and NAD⁺ were enhanced, and the affinity of NAD⁺ showed a better binding advantage (Supplementary Table 2). The $K_m$ of E4P of Gni_Epd and Eco_Epd are 0.737 and 0.969 mM, respectively. The $K_m$ of NAD⁺ of Gni_Epd and Eco_Epd are 0.051 and 0.717 mM, respectively (Table 1). Notably, despite a reduction in protein expression (Supplementary Fig. 3c), the catalytic efficiency of Gni_Epd was significantly enhanced (Table 1), particularly for the substrate E4P (with an almost 3-fold improvement in $k_{cat}/K_m$). This suggests that the increase in product yield was primarily attributed to the improved catalytic capacity of Gni_Epd, as well as its enhanced binding and recruitment of E4P and NAD⁺. Specifically, the binding energies were calculated for the tested species (discussed in detail in Supplementary Discussion 1). Furthermore, we discovered that the measured $K_m$ values are consistent with those reported in previous studies[59,60], while the $k_{cat}$ values differ significantly from those previous findings (this study $K_m = 0.969$ mM, $k_{cat} = 0.15$ s⁻¹, previous studies $K_m = 0.510$–0.960 mM, $k_{cat} = 20$–200 s⁻¹). The experimental data and measurement methods were thoroughly reviewed and validated to ensure the accuracy and reliability of the data. Based on the obtained data, we can suggest that the binding and catalysis of Epd may represent a potential bottleneck in the pathway. Therefore, further optimization of $k_{cat}$ and the binding interactions of E4P and NAD⁺ could effectively enhance the substrate flow toward the PN pathway and further improve PN production.

DXP is generated from pyruvate and G3P in a thiamine diphosphate (ThDP)-dependent reaction catalyzed by Dxs (Fig. 1). It is required for

the biosynthesis of PN and isoprenoid compounds, while also being a precursor of ThDP, which is required for the formation of DXP itself[66]. The availability of DXP is key to improving PN yields, as Dxs catalyzes a flux-controlling step[53]. To identify suitable Dxs heterologous enzymes for PN production, the BRENDA enzyme database (https://www.brenda-enzymes.org/) was searched. Several Dxs heterologous enzymes which have higher turnover numbers or $k_{cat}/K_m$ values in the target reaction compared to Eco_Dxs were selected, including *D. radiodurans*, *A. tumefaciens*, and *R. capsulatus*. Additionally, the commonly used genes in biomanufacturing were also selected to study the *dxs* expression effects derived from *R. sphaeroides*[67–69] and *B. subtilis*[70–72] on PN production in *E. coli*. Notably, two heterologous *dxs* genes from *A. tumefaciens* due to their relatively high turnover numbers and the *dxs* gene from *E. meliloti*, a natural overproducer of vitamin B₆, were further studied (Supplementary Table 3). The *dxs* genes were overexpressed using the p15ASI-*epd* backbone, which confirmed that the *dxs* gene from *E. meliloti* has a great promotion effect on the production of PN, with a titer of 68.14 mg/L, representing a 6.74-fold increase compared with the native gene (Fig. 4d). As this may be related to the stability of the protein, we plotted the stability map (Supplementary Fig. 7) of the structures predicted by AlphaFold2 using the CNA web server. Eme_Dxs exhibited more rigid links than Eco_Dxs, which is beneficial for protein stability, and may, in turn, increase the protein expression to enhance the PN yield. In conclusion, Eme_Dxs was identified as a better enzyme for the production of PN under the tested conditions.

## Expanding intermediate metabolic pathways through PdxB and SerC metabolic engineering

A series of "sluggish" enzymes catalyze the formation of vitamin B₆ because it is required in very small amounts in the cells[1,26]. To further

**Table 1 | Kinetic parameters of purified Eco_Epd and Gni_Epd toward E4P and NAD$^+$**

| Epd | E4P | | | NAD$^+$ | | |
|---|---|---|---|---|---|---|
| | $K_m$ (mM) | $k_{cat}$ (s$^{-1}$) | $k_{cat}/K_m$ (s$^{-1}$/mM) | $K_m$ (mM) | $k_{cat}$ (s$^{-1}$) | $k_{cat}/K_m$ (s$^{-1}$/mM) |
| Eco_Epd | 0.969 | 0.150 | 0.155 | 0.717 | 3.232 | 4.508 |
| Gni_Epd | 0.737 | 0.323 | 0.438 | 0.051 | 0.313 | 6.137 |

Source data are provided as a Source Data file.

increase the vitamin B$_6$ yield, we engineered PdxB and SerC to increase the metabolic flow of the intermediate metabolic pathways.

PdxB, which is a nicotino-enzyme wherein the NADH cofactor remains tightly bound to the protein, catalyzes the second step of vitamin B$_6$ biosynthesis[73,74]. The catalytic constant of PdxB is the best among the enzymes in the vitamin B$_6$ biosynthesis pathway[13]. A previous study reported that a variety of physiologically available α-keto acids serve as oxidants of PdxB to sustain multiple turnovers[73]. α-Ketoglutarate (α-KG) can serve as a physiological reoxidant of PdxB and is also a product of SerC aminotransferase[73]. We, therefore, tested whether the addition of α-KG (1, 5, 10, 20 mM) can enhance PN production. We used the test strain LL370, in which PdxB was overexpressed downstream of Gni_Epd and formed an operon based on the LL239 mutant. As shown in Fig. 5a, the exogenous addition of α-KG promoted PN production in the range of 5–20 mM, with a slight positive effect on growth observed. Furthermore, the ratio of PN production to OD$_{600}$ was found to be higher in the presence of 5, 10, or 20 mM of α-KG compared to the blank control. These results suggest that the exogenous addition of α-KG can be effective in promoting the production of PN. In the biosynthesis pathway of vitamin B$_6$, it can be noted that SerC-catalyzed transamination reaction leads to the production of α-KG (Fig. 1). Thus, the overexpression of SerC in close proximity to PdxB can offer a dual benefit of increasing the metabolic flux of PN biosynthesis while also aiding in the multiple turnovers of PdxB through spatial proximity.

The downstream enzyme of PdxB is SerC, a PLP-dependent aminotransferase that plays roles in the biosynthesis of PN, serine, and lysine by using different substrates[75,76]. The redundancy and promiscuity of SerC have been widely studied[77–80]. As simulated by iML1515[27], the growth rate of *E. coli* was 0.49 h$^{-1}$ when the uptake rate of glycerol was set to 10 mmol gDCW$^{-1}$ h$^{-1}$. The SerC enzyme catalyzes three reactions related to the metabolism of vitamin B$_6$, serine, and lysine, and the corresponding fluxes were 0.0001, 0.8497, and 0.1835 mmol gDCW$^{-1}$ h$^{-1}$, respectively (Fig. 5b). To improve the metabolic flux of vitamin B$_6$, the D-3-phosphoglycerate dehydrogenase SerA was knocked out in LL006 to inhibit the production of the competing substrate 3-phosphooxypyruvate in the serine synthesis pathway. Compared with the WT, the *ΔserA* strain indeed had a 223% higher PN yield, but its biomass was reduced by 48% after 24 h (Fig. 5c). It was reported that high concentrations of serine inhibit the growth of *E. coli*[81]. Although different concentrations (2, 5, 10, 50, 100 mM) of serine were added to the medium, the growth of the *ΔserA* strain was not restored to a level comparable to the starting strain *E. coli* (Fig. 5d). We also supplemented the medium with 2, 5, 10 mM glycine, which can be converted to serine but is less toxic[81]. The results indicate that glycine promotes better growth than serine and the toxic effect of serine addition has been shown at the lowest concentration of 2 mM ($p = 0.0248$, Student's two-tailed $t$-test) compared with the addition of the same concentration of glycine in the tested condition at 24 h. Additionally, the growth was found to be almost completely restored when 10 mM glycine was added (Fig. 5d). Although the *ΔserA* mutant strain showed an increase in the productivity of PN, the severe growth defects would result in longer production cycles and may cause difficulties in large-scale production. To overcome the growth defect issue, *serC* was overexpressed to increase the biosynthetic flux in the PN pathway.

## Iterative multimodule optimization strategy to boost the production of PN

Modular metabolic engineering enables the global fine-tuning of the expression levels of engineered pathways by modularizing the synthetic pathway to reconstitute the metabolic balance in the hosts[54,82,83]. In this study, the overall vitamin B$_6$ pathway was partitioned into an upstream push module and a downstream pull module (Fig. 6a). The modules were separated at 4HTP, which is a toxic intermediate. The upstream module consists of Epd, Dxs, PdxB, and SerC to enhance the push driving force by providing sufficient DXP and the intermediate 4HTP. The downstream pathway consists of PdxA and PdxJ to increase the pull force (Fig. 6a).

The effect of vector copy number on the PN yield was evaluated and the results showed that the push module was optimally expressed from the low-copy plasmid p15ASI, while the medium-high copy plasmid pRSFDuet-1 was better suited for the pull module. We speculated that the upstream module required low expression because the intermediate 4HTP is toxic. The first round of operon-optimization yielded the strain LL372, which overexpressed codon-optimized *pdxA* and the original *pdxJ* sequence. Gni_Epd and Eme_Dxs were selected to build the upstream module. The genes encoding the upstream enzymes were sequentially inserted into the p15ASI plasmid one by one, and each subsequent enzyme-coding gene increased the fermentation yield following optional RBS optimization (Fig. 6b and Supplementary Fig. 8). Considering that SerC is involved in multiple biosynthetic pathways, using the strong promoter J23119 to control the overexpression of SerC resulted in a better PN yield. It was suggested that the cell needs enough SerC protein to guide metabolic flux to the PN biosynthesis pathway. Additionally, the match of the upstream module revealed a highly nonlinear PN flux landscape with a global maximum resulting in a 247.78-fold increase of vitamin B$_6$ production over the starting strain (from 0.42 to 104.07 mg/L PN) (Fig. 6b). Here, the final upstream vector [p15ASI-P$_{tac}$-*epd* (Gni)-*pdxB* (Eco)-*dxs* (Eme)-P$_{J23119}$-*serC* (Eco)] was named R42.

Next, the second round of iterative multimodule optimization was performed on the downstream module to increase the pull driving force. This round of operon optimization was based on the strain LL372b, which overexpresses the upstream module from the R42 plasmid. The downstream module is important to pull the flux to produce the first B$_6$ vitamer PNP. The expression of the *pdxA-pdxJ* operon was driven by the strong constitutive promoter J23119. We first modulated the strength of the promoters using publicly available online tools (https://www.denovodna.com). Considering that the J23119 promoter may be too strong to fold the protein properly, two promoters with lower transcription initiation rate (TIR, au) (P2/P3) compared to the J23119 promoter (P1) were selected (Supplementary Table 4). As shown in Fig. 6c, promoters with low transcriptional intensity generally increased the PN production, but it did not increase when we used a promoter with a lower intensity than P3, so the strength of the P3 promoter may be more suitable for the expression of *pdxA* and *pdxJ* for PN production (Fig. 6c). The results showed that suitably low transcription level of *pdxA* and *pdxJ* may contribute to correct protein folding, but too low expression would affect the amount of enzyme, thus reducing the yield.

In addition, two mini CDS libraries of PdxJ and PdxA were used to further balance the metabolic flow. We chose PdxJ mutants with better performance, including PdxJ1 (E104T/I218L/G194C), PdxJ2 (I13V/L212I/E104T), PdxJ3 (V44I/L212I/A23T), PdxJ4 (I10L/T22V), PdxJ5 (G194C/A23T/I219G), PdxJ6 (V44I/L212I/A23T/V101L), etc., to identify the variant with better yield. Two controls were set up in the left panel of Fig. 6d. Compared to LL372b, the downstream module played a great pulling role after the overexpression of natural PdxJ and its mutants. In addition, the mutants further increased the pulling force compared with LL379, prompting an increase in yield. The results showed that PdxJ1 displayed the best performance, enhancing the PN titer to

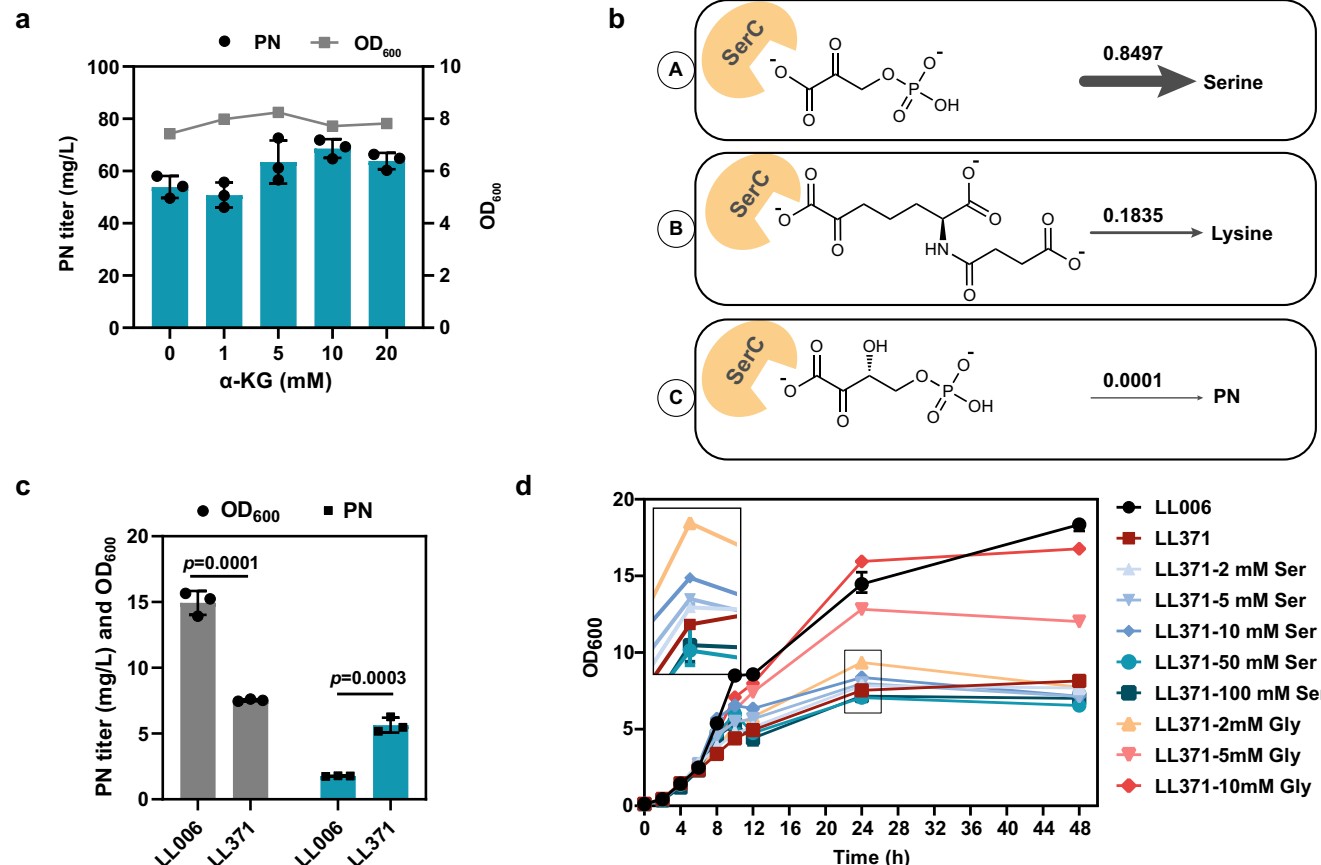

**Fig. 5 | Metabolic engineering of PdxB and SerC. a** The PN titer and cell growth of LL239 with or without α-KG addition in 24 deep well plates. LL239: LL006 harboring pRSFDuet-1_*pdxJ* (E104T/I218L/G194C). **b** The SerC enzyme catalyzes three reactions related to the metabolism of serine, lysine and PN. The substrates were 3-phosphonooxypyruvate (A), N-succinyl-2-L-amino-6-oxoheptanedioate (B), and OHPB (C), respectively. The corresponding fluxes are displayed in numbers and the thickness of arrows represents the proportion of carbon flow. **c** The PN titer and OD$_{600}$ of the control strain (LL006) and the *serA* deletion mutant LL371 (LL006

*ΔserA*) at 24 h. Effect of deleting SerA on growth and PN titer in FM1.4 medium. **d** The growth curves of the control strain (LL006) and *E. coli* LL371 (LL006 *ΔserA*) in the fermentation medium with or without serine (Ser) and glycine (Gly) at 37 °C. The image shown at the top-left corner represents an enlarged view of the OD$_{600}$ measurements taken at 24 h. **a**, **c**, **d** Experiments were conducted in triplicates (*n* = 3), and measurements are represented with their means and s.d. Significance (*p*-value) was evaluated by two-sided *t*-test. Source data are provided as a Source Data file.

124.05 mg/L (Fig. 6d). Similar strategies were used to optimize the fitness of PdxA mutants, including PdxA1 (P245C), PdxA2 (H136N), PdxA3 (V149L) and PdxA4 (E214N), using PdxJ1 (E104T/I218L/G194C) as a control (Fig. 6d). The results showed that PdxA2 (H136N) was the best PdxA mutant, and when expressed under the control of the P3 promoter, it enhanced the PN titer to 453.78 mg/L in shake flasks (Fig. 6d). These results suggested that the effective performance of the downstream driving force requires the fitness of multiple enzymes. Additionally, the growth was not affected by the expression plasmid (Fig. 6d). The best strain LL388, overexpressed PdxJ1 and PdxA2 under the P3 promoter (P3A2J1), produced 48.86-fold more PN than the strain expressing only the native upstream pathway (LL372b, 9.10 mg/L). The construction of mini CDS libraries showed that changes in expression levels in the downstream pathway resulted in greatly improved PN accumulation, which may be attributed to the enhanced pull of metabolic flux and the improved fitness between modules to promote PN biosynthesis.

To study phenotypic and genetic diversities in PN-producing *E. coli* populations, the starting strain LL388 was tested by tracking growth and PN production through serial passaging[84,85]. To prevent plasmid loss, we kept the cultures at a constant antibiotic selection. By serially transferring the *E. coli* every 12 h, we achieved populations >100 cell generations. During the 40th–100th generations, a significant decrease in PN yield was observed, and ~15% of the cells lost the

plasmids (Supplementary Fig. 9a and b). Additionally, we observed changes in colony morphology, with a fraction of colonies (~15%) exhibiting larger size and thinner edges, suggesting the presence of mutants that had lost the plasmids (Supplementary Fig. 9c). However, there were no significant differences observed in the growth curves among the various generations, although the 60th and 100th generation cells exhibited slightly faster growth compared to the parental strain (1st generation) (Supplementary Fig. 9d). During the small-scale fermentation with shake flask cultivation for up to 48 h, the cells that contained the plasmids were found to be consistently maintained at a level of 100%. Thus, the development of plasmid with increased stability or plasmid-free strains over timescales relevant to industrial levels will further improve PN production.

## Fed-batch fermentation

Fed-batch fermentation of the vitamin B$_6$ high-producing strain LL388 was performed in a 5 L bioreactor. In this process, NH$_4$OH was used as a neutralizer to control the pH at 6.5. The initial concentration of glycerol was 15 g/L. In the process of fed-batch fermentation, the dissolved oxygen (DO) was maintained at 30% by adjusting the agitation rate. After 48 h, 1409 mg/L PN was produced, which was 3.1-fold higher than in shake flask culture using FM1.4 medium and equivalent to a productivity of 29.16 mg/L/h. Compared with the previously reported *E. coli* strain, which produced mixed B$_6$ vitamers as a measurement, the

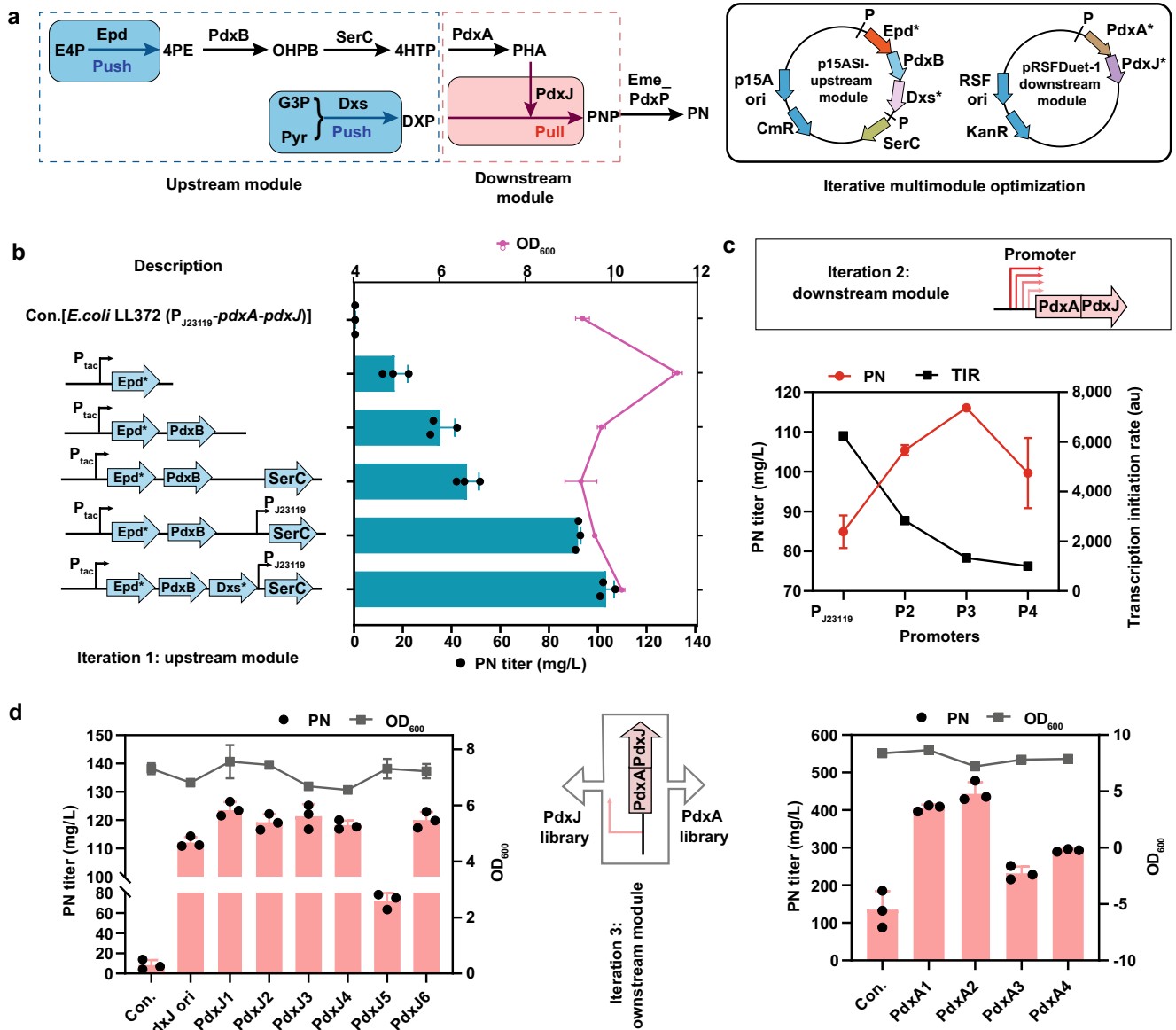

**Fig. 6 | Iterative multimodule optimization of the vitamin B_6 biosynthetic pathway. a** Scheme of the up- and downstream modules and the vectors used in the iterative multimodule optimization. **b** The first round of iterative multimodule optimization of the upstream module. The control (Con.) is the LL372 mutant. **c** The second-round promoter optimization of the downstream module. **d** The third-round iterative multimodule optimization of the CDS (coding sequences) library of PdxA and PdxJ. The control (Con.) of the left panel is the mutant LL372b, which only contains the R42 plasmid of the upstream push module. "PdxJ ori" is the LL379 (LL372b harboring pRSFDuet-1-P3-*pdxA-pdxJ*). The control (Con.) of the right panel was the mutant LL381, which contains the R42 [p15ASI-P_tac-*epd* (Gni)-*pdxB* (Eco)-*dxs* (Eme)-P_J23119-*serC* (Eco)] and the pRSFDuet-1-P3-*pdxA-pdxJ*1 plasmids. **b**–**d** Data are presented as mean values ± SD from three independent biological replicates (*n* = 3), the circles represent individual data points. Source data are provided as a Source Data file.

engineered strain LL388 exhibited an 18-fold increase in PN production[23]. As shown in Fig. 7, the maximum OD_600 of LL388 only reached 44.8, which may be constrained by the metabolic burden from plasmid maintenance and heterologous protein expression or the scale-up fermentation process. Thus, strain improvement by genetic strategies such as constructing a plasmid-free strain and the optimization of the fermentation process will be further studied to enhance cell growth and increase PN production.

## Discussion

Green biomanufacturing, regarded as the most promising production model to replace the unsustainable fossil economy, uses renewable resources as raw materials and constructs efficient biological processes for the production of fuels and chemicals[86,87]. Vitamin B_6 is an

essential nutrient that is highly demanded by pharmaceutical, feed, and food processing markets. Here, we decoupled PN production from cell growth by orthogonal twin-pathway engineering in *E. coli*. Moreover, two libraries with more than 350 site-directed mutations were constructed through rational engineering, and 18 heterologous genes in silico were screened to find improved enzyme variants better suited for PN production. Finally, an iterative multimodule optimization strategy was successfully implemented to achieve a PN titer of 1.4 g/L with productivity of 29.16 mg/L/h.

Growth decoupling is a promising strategy that is mainly based on carbon source switch or metabolic shift[35,88,89]. In our study, the synthesis of PLP required for cell growth and PN production were decoupled and modulated to maintain cell growth and enhance PN production using two parallel biosynthetic pathways (Fig. 2b and c). The

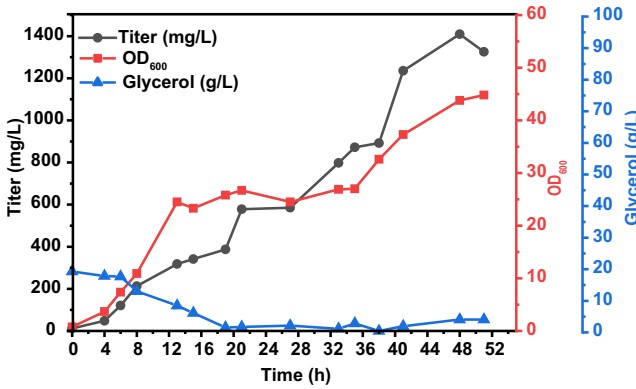

**Fig. 7 | Biosynthesis of PN by fed-batch fermentation of the LL388 strain in a 5 L bioreactor.** The glycerol concentrations, PN titer, and OD$_{600}$ were recorded during the whole fermentation process. The black line indicates the PN titer, the red line shows the trend of OD$_{600}$, and the blue line represents the glycerol concentration. Source data are provided as a Source Data file.

overexpression of a PdxP homolog identified from *E. meliloti*[39] was also critical for an engineered strain to achieve a high PN production potential since the PN production pathway was complete. Introduction of the R42 and P3A2J1 plasmids (Fig. 6) into WT *E. coli* or the engineered LL005 strain resulted in much lower PN yields than when using LL006 as chassis cells (Supplementary Fig. 10). These results illustrate the importance of decoupling PN production from cell growth in parallel pathway engineering and constructing a complete pathway by introducing PNP phosphatase.

In general, protein engineering has yielded positive results when applied to various enzymes[90,91]. Combining multiple enzymes with high catalytic efficiency does not always result in a synergistic effect due to the complexity of biological systems, but the high catalytic efficiency of each enzyme is necessary for a robust pathway. Therefore, we implemented protein engineering modifications such as sequence-based screening and structure-based computation design throughout the DXP pathway to ensure a robust pathway was established. Experimental data has demonstrated that protein engineering can significantly enhance the production of PN. Specifically, Epd homolog mining increased yield by 2.2 times, Dxs by 6.75 times, and PdxA by 1.6 times. Notably, PdxJ resulted in an 18-fold increase in the PN titer compared to native *E. coli* enzymes (Figs. 3 and 4). To reduce the experimental workload, we employed a parallel summation scheme instead of an exponentially increasing combinatorial scheme to generate libraries of protein mutants or homologs at each step in the pathway. This ensured that every gene had a positive additive effect.

Additionally, push and pull driving forces play important roles in the biosynthesis of target compounds[56,92,93]. The 'push-pull' approach combined with "restrain" or "block" strategies have been successfully applied to regulate the modular expression of distinct sets of genes in metabolic pathways[55,56,93–96]. In this context, the iterative multimodule optimization strategy based on the push-pull concept was used to boost PN production. The advantage of this strategy is that the stacking of each step increases the PN yield, using either promoter, RBS, or CDS libraries. By enzyme superposition and promoter regulation, the upstream module pushed more metabolic flux into the PN biosynthesis pathway. The downstream module can be enhanced first by downregulation of the initial transcription rate, which may be helpful to reduce the burden on the host. PN production can be further increased by a library containing beneficial mutants screened using protein engineering approaches. PN production was enhanced with little variation among *pdxJ* mutants and *pdxA* mutants, but it resulted in a large promotion compared with the starting strains (Fig. 6d).

Overexpression of the up- or downstream module alone could slightly increase the PN titer, but they could significantly boost the titer when overexpressed in combination (approximately 48.86-fold increase if downstream module expression in the strain included the upstream module, and 247.78-fold conversely). This suggests that the enhanced pull force of the downstream module and the improved catalytic efficiency of the upstream module contributed to a more efficient conversion of the substrate into the target compound.

Further studies can be performed to improve PN through multi-omics studies. Based on the cellular mounts of the overexpressed proteins by proteome investigations (Supplementary Method 1), it was evident that the Eme_Dxs and SerC proteins may be the potential bottlenecks in the DXP-dependent vitamin B$_6$ pathway due to their lower expression ratio compared to other proteins (Supplementary Table 5). Especially, the Eme_Dxs exhibited lower expression during the fed-batch process compared to shake flask fermentation (Supplementary Table 5). The observed decrease in Dxs protein expression might result in a shortage of the precursor DXP, potentially limiting both PN production and cell growth on a larger scale. Therefore, it is crucial to ensure adequate expression of Eme_Dxs to achieve optimal PN production and promote cell growth in larger quantities. Additionally, the main considerations of future experiments include in-depth multi-omics studies, the construction of a plasmid-free strain, cofactor engineering, and the scale-up of the fermentation process. All these factors substantially affect the final production of bioprocess and therefore need to be systematically examined.

In conclusion, the engineered *E. coli* strain is able to efficiently produce PN with titer 453.78 mg/L of PN in a shake flask and 1409 mg L$^{-1}$ in a 5 L fed-batch fermentation with productivity of 29.16 mg/L/h. It is expected that the production of PN will be further increased by the measurement of intermediate concentrations to uncover remaining bottlenecks, followed by detailed optimization of fermentation conditions. The iterative multimodule optimization strategy provides a good reference for improving the production of other valuable chemicals. In addition, our study lays a solid foundation for the green bioproduction of vitamin B$_6$ and its future industrialization.

## Methods

### Strains and growth conditions

All engineered *E. coli* strains used and constructed in this work are derivatives of the WT strain *E. coli* (MG1655). The *E. coli* DH5α strain was used for plasmid construction. All strains, primers, and heterologous or codon-optimized gene sequences used in this work are listed in Supplementary Data 4–6. All plasmids are listed in Supplementary Table 6. *E. coli* was grown in LB medium, and transformants were selected on plates containing kanamycin (50 μg/mL), or chloramphenicol (34 μg/mL), as appropriate.

For fermentation, activated colonies from the frozen stock were inoculated into tubes containing 5 ml of seed medium: 10 g/L glycerol, 10 g/L Bacto peptone, 5 g/L yeast extract, and 5 g/L NaCl (pH not adjusted) containing the appropriate antibiotics. After shaking the tubes for 16 h at 37 °C, the culture was transferred into 24 deep-well plates or flasks with FM1.4 medium (15 g/L glycerol, 1 g/L glucose, 10 g/L Bacto peptone, 5 g/L yeast extract, 200 mg/L MgSO$_4$·7H$_2$O, 10 mg/L FeSO$_4$·7H$_2$O, 10 mg/L MnSO$_4$·5H$_2$O, 100 mM Na$_2$HPO$_4$, pH 6.5) with the appropriate antibiotics.

### Computational enzyme redesign

The crystal structures of PdxA (PDB 1PS6) and PdxJ (PDB 1M5W) were extracted from Protein Data Bank[97]. Protein Preparation Wizard (PrepWizard)[98] was applied to prepare the protein structure in Schrödinger Maestro 2018 (Schrödinger, Llc, New York, USA). Then, the protein structure was energy-minimized using the OPLS3 force

field[99]. The transition state analog (TSA) was constructed based on catalytic mechanisms for enzyme design[45].

The Glide module[100] of Schrödinger 2018 was employed to generate the TSA. After protein preparation and minimization, TSA substrates were docked into the binding pocket of each protein. The grids were generated at the centroid of the ligands in the crystal structures. Standard precision was used for the docking methods. Default settings were employed for other parameters for the grid generation. Structures that met the catalytic conditions and had the lowest docking scores were used for the design of the simulated mutations.

For each TSA structure, residues approximately 6 Å from the TSA were redesigned to evaluate the binding energy using the Rosetta Enzyme Design application. The command line parameters -enzdes -detect_design_interface -cut1 0.0 -cut2 0.0 -cut3 10.0 -cut4 12.0 -cst_opt -chi_min -bb_min -cst_min -cst_design -design_min_cycles 2 -lig_packer_weight 1.8 -packing:use_input_sc -packing:soft_rep_design -packing:linmen_li 10 -nstruct 200 were applied and written in a "flag-file". The Rosetta Enzyme Design application optimizes the catalytic location of the substrate by applying forces between the bound substrate and the important residues. The geometry of the TSA conforms to the transition state and is based on the catalytic mechanism and the crystal structure. The Rosetta Enzyme Design application utilizes a Monte Carlo algorithm to select mutations and structural changes that reduce the overall energy to generate the redesigned 3D structure. Multiple rounds of designs were performed based on the experimental results. For each round of design, saturating mutagenesis was performed on individually selected residues, after which the mutation with the best binding energy was selected for experimental validation.

## Molecular dynamics (MD) simulations

All TSA complexes were based on Rosetta Design results. AMBER18[101] was used for energy minimization of the constructed models and MD simulation of all the models was conducted using the ff14SB force field. The constrained MD simulations were conducted for 20 ns. Finally, three independent 100 ns MD simulations were performed without any restriction. The complete simulation methodology used in this work is available in Supplementary Method 2.

## DNA manipulation and site-directed mutagenesis

The primers used in this study were synthesized by GENEWIZ Biotechnology Co., Ltd. (Suzhou, China). All recombinant plasmids were constructed using the DNA seamless cloning technology according to the instruction of ClonExpress® MultiS One Step Cloning Kit (Vazyme Biotech Co., Ltd.; catalog number C113-01). Gene knockout and substitution in *E. coli* were performed using the CRISPR-Cas9 system for scarless genomic editing based on the pRed_Cas9_recA plasmid[38]. A synthetic ribosome binding site (RBS) was generated to drive protein translation at a specified predicted rate using the RBS Calculator online tool (https://www.denovodna.com/)[102].

All heterologous genes and partial endogenous genes, such as *epd* (GenBank accession AAC75964.1) and *pdxA* (GenBank accession AAC73163.1), from *E. coli* were codon-optimized according to the *E. coli* codon preference, and other endogenous genes were amplified from the *E. coli* genome. The codon-optimized sequences were synthesized by GENEWIZ and then cloned into p15ASI or pRSFDuet-1. The *pdxJ* gene (GenBank accession AAC75617.1) was directly cloned from *E. coli*. The inducible T7 promoter together with the *lac* operator, and RBS sequences in pRSFDuet-1 was replaced by the constitutively expressed promoter J23119 (lately optimized) and the synthetic RBS sequences to express *pdxA* and *pdxJ* genes. Point mutations to change the amino acid sequence were introduced by polymerase chain reaction (PCR) [Takara PrimeSTAR® Max DNA Polymerase (catalog number R045Q), direct and reverse primers 10 μM, template plasmid <100 ng] with plasmid pRSFDuet-1_*pdxA* or *pdxJ* used as the template. The PCR products were treated with FD DpnI at 37 °C in a heat block for 5 min to remove the intact template plasmids before being transformed into DH5α competent *E. coli*. The resulting mutated plasmids were further transformed into the engineered strain. The primers were designed using CE Design software from Vazyme Biotech Co., Ltd.

## In vitro enzyme assays

The plasmids pET-Eco_epd and pET-Gni_epd were transformed separately into *E. coli* BL21 (DE3), and the resulting recombinant strains were cultured until they reached an $OD_{600}$ of 0.6 prior to induction with 0.5 mM IPTG for 12 h at 25 °C. Cells were subsequently harvested and re-suspended in lysis buffer (50 mM Tris–HCl, 200 mM sodium chloride, 20 mM imidazole, pH 8.6). The His-tagged proteins were purified using $Ni^+$-affinity chromatography and protein concentration was determined using the Bradford Protein Assay Kit (Beijing Solarbio Science & Technology Co., Ltd., catalog number PC0010).

Kinetic analyses were performed on purified Eco_Epd and Gni_Epd by incubating the enzyme with reaction mixtures (0.2 mL) comprising of 50 mM PPi buffer (pH 8.6), 1 mM DTT, varying concentrations of $NAD^+$ or E4P, and 0.01–0.1 μg of a pure enzyme. The kinetic parameters were determined by fixing one substrate concentration at saturation ($NAD^+$ = 2 mM, or E4P = 8 mM) while varying the concentration of the other substrate (E4P from 0.05 to 8 mM, or $NAD^+$ from 0.01 to 2 mM). Increases in A340 were monitored for 3 min at 37 °C to measure the rate of the enzyme reaction[59,60].

## Small-scale PN production assay and fed-batch fermentation

To monitor PN production, we inoculated the precultures with single colonies from glycerol stocks grown on LB agar plates. The precultures in 5 mL of inoculated seed medium were incubated overnight at 37 °C and 220 rpm, and then used to inoculate 24 deep-well plates containing 2 mL FM1.4 medium to a final $OD_{600}$ of 0.1. The plates were incubated in a high-speed shaking incubator (Zhichu, China) at 37 °C, 800 rpm, and 80% humidity for 48 h. The cells of 1 ml samples were removed by centrifugation at 6200×*g* for 5 min, and the supernatants were used for HPLC analysis of PN production. Shake-flask fermentation was similar to fermentation in 24 deep-well plates. The seed cultures were transferred into 250 mL shake flasks, each containing 30 mL of culture medium, and adjusted to an initial $OD_{600}$ of 0.1, followed by incubation for 48 h at 37 °C with shaking at 200 rpm. All cultivations were performed in duplicates or triplicates.

The fed-batch culture was performed in a 5 L bioreactor with a 2 L working volume. A fresh single colony of the constructed derivatives was seeded into 5 mL of LB medium and cultured at 37 °C. Subsequently, 1% of the culture was transferred into 200 mL of seed medium in 1000 mL shake flasks and cultivated at 37 °C for 16 h. Finally, the seed culture was used to inoculate 2 L of fermentation medium at a 10% inoculation percentage. The DO level was controlled at 30% air saturation by coupling varying the agitation speed from 300 to 800 rpm. The pH was kept at 6.50 via the automatic addition of $NH_4OH$ (25%, v/v) and $H_3PO_4$ (20%, v/v). The glycerol concentration was maintained below 5 g/L by feeding 450 g/L glycerol, 5 g/L yeast extract, 6 g/L $MgSO_4 \cdot 7H_2O$, 0.3 g/L $FeSO_4 \cdot 7H_2O$, and 0.3 g/L $MnSO_4 \cdot 5H_2O$ in the fermentation process.

## Analytical methods

The cell density ($OD_{600}$) was measured using a Hybrid Multi-Mode Reader (Synergy Neo2, Bio Tek, USA). PN was quantified using a ThermoFisher high-performance liquid chromatography (HPLC) system (UltiMate™ 3000) equipped with an FLD-3400 detector using a gradient program according to previously published protocols with minor modifications as follows[103]. Briefly, the fermentation mixture was centrifuged, and the resulting supernatant was used for HPLC analysis employing fluorescence detection. The excitation and emission wavelengths were 293 and 395 nm, respectively. PN was separated by the use of an octadecylsilyl (ODS) column (Cosmosil AR-II; 250 by

4.6 mm, 5 μm particle size; Nacalai Tesque) using a gradient program. Mobile phase A (33 mM phosphoric acid and 8 mM 1-octanesulfonic acid, adjusted to pH 2.4 with KOH) and mobile phase B (80% acetonitrile [vol/vol]) were used for separation. The total flow rate was 0.8 ml/min. The gradient program (linear gradient) was as follows: 0% mobile phase B to 1% B for 5 min, 1% B to 19% B for 5 min, 19% B to 28% B for 10 min, 28% B to 63% B for 5 min, 63% B to 0% B for 2 min, and 0% B for 5 min. All data are expressed as the means ± SD from three independent replicates. The data from fermentation and enzyme assays were analyzed by the software Origin Pro (version 9.1) or GraphPad Prism (version 8).

PNP and PLP were detected using an Agilent 1260 HPLC system connected to a Bruker micrOTOF-Q II mass spectrometer. Except for the different mobile phases (A: formic acid in water, pH = 2.5 and B: 100% methanol) used[104], other details are the same as HPLC. MicrOTOF and DataAnalysis were used to collect and analyze the HPLC-MS data, respectively.

### Sequence analysis and phylogenetic tree construction
The protein sequences of Epd were extracted from the BRENDA database based on EC number 1.2.1.72. Multiple sequence alignment and comparative analysis of the protein were conducted using the command-line version ClustalW2[105]. Phylogenetic trees were constructed using the data from the alignments in MEGA11 software[63]. Protein structure analysis and figure design were performed using PYMOL 2.1.

### Genetic stability assay
The genetic stability of the engineered strain was carried out at different stages of cultivation, i.e. seed (at 12 h) and serially transferred at 12 h intervals to FM1.4 production medium. The sample of seed (1st), 60th, and 100th consecutive generations were plated on nonselective plates and incubated at 37 °C for 16–20 h. Single colony was randomly picked and spotted on LB plates with and without kanamycin and chloramphenicol. The colonies were counted and the ratio of the number of colonies on plates with and without antibiotics was used to calculate percentage stability. The colony morphology was been observed by Leica stereo microscopes (M205C).

### Metabolic flux analysis for vitamin B$_6$ biosynthesis
The optimal growth rate and flux distribution for the biomass objective function were calculated using the genome-scale metabolic model iML1515[27] of *E. coli* by conducting flux balance analysis (FBA)[106]. The COBRApy toolbox was used to perform FBA in Python[107], and CPLEX (IBM, Armonk, NY, USA) was used as the linear programming solver. For the simulation, glycerol was the only carbon source, and the uptake rate was set to 10 mmol gDCW$^{-1}$ h$^{-1}$. The sources of nitrogen, oxygen, phosphate, and sulfur were not constrained. The three reactions catalyzed by SerC are 'OHPBAT', 'PSERT', and 'SDPTA', which are related to the metabolism of vitamin B$_6$, serine, and lysine.

### Reporting summary
Further information on research design is available in the Nature Portfolio Reporting Summary linked to this article.

## Data availability
The raw proteomic data have been deposited in Zenodo [https://doi.org/10.5281/zenodo.8248148]. Sequence data in this article can be found in Supplementary Data 6. Source data are provided with this paper.

## Code availability
Protein stability design codes are publicly available at Github [https://github.com/Gerald-li/Protein_Stability_design-workflow] or Zenodo [https://doi.org/10.5281/zenodo.8198581].

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

## Acknowledgements

This work was supported by the National Key R&D Program of China (2022YFC2106100 to L. L.), the National Science Fund for Distinguished Young Scholars (22325807 to D.Z.), the National Natural Science Foundation of China (32200049 to L.L., 22178372 to D.Z.), Tianjin Synthetic Biotechnology Innovation Capacity Improvement Project (TSBICIP-CXRC-004 to L.L., TSBICIP-CXRC-055 to D.Z., TSBICIP-PTJJ-007 to D.Z.), TianHe Qingsuo Open research fund of TSYS in 2022 & NSCC-TJ (P-THQS-22-ZD-No.0003 to J.L.), and the Development of platform technologies of microbial cell factories for the next-generation biorefineries project (2022M3J5A1056117 to S.Y.L.) by the Korean Ministry of Science and ICT through the National Research Foundation.

## Author contributions

L.L.: conceptualization, methodology, validation, investigation, visualization, writing—original draft, funding acquisition. J.L.: software, visualization, writing—original draft. Y.G.: conceptualization, methodology, validation. Y.W., Z.T., and T.W.: investigation, resources. P.L., Q.Y., and H.M.: software, writing—original draft. S.Y.L. and D.Z.: conceptualization, supervision, funding acquisition, writing—reviewing, and editing.

## Competing interests

The authors declare no competing interests.
