## [Peer Review File · Nature Communications]

Protein engineering and iterative multimodule optimization for vitamin B6 production in Escherichia coliReviewers' Comments:

Reviewer #1:

Remarks to the Author:

With great interest I was reading the manuscript by Liu and co-worker about the engineering attempts of *Escherichia coli* for overproducing the commercially attractive B6 vitamer pyridoxine (PN). For this purpose, various genetic modules were constructed to overexpress the genes of the DXP-dependent vitamin B6 pathway in *E. coli*. Moreover, the *pdxST* operon from *Bacillus subtilis* was co-expressed in the background of a *pdxH* deletion mutant. By following this approach, the authors managed to produce 1.4 g/L of PN, which is the highest reported vitamin B6 titer. The amount of work to enhance the production of PN is quite large. However, I have several issues that need to be addressed by the authors.

Major points

1. Previously, it has been shown that the *pdxST* genes from *B. subtilis* can be functionally expressed in *E. coli* for producing the biologically active B6 vitamer PLP. Therefore, it is unclear why the authors tested the *pdxST* genes from different organisms. Some of the data presented in Figure 2 can be removed.
2. In Figure 2b it is shown that the B6 vitamer PN complements PLP auxotrophy of the *E. coli* *pdxH* mutant. How is this possible? The *pdxH* mutant cannot convert PNP to the biologically active B6 vitamer PLP. How much PN was added to the medium? Usually, commercially available PN contains a 1000-fold lower amount of PL. The *pdxH* mutant can easily be complemented with PL.
3. Pages 3-4, lines 44 – 70: The introduction or at least the red line of the introductory text is relatively close to existing reviews about vitamin B6 metabolism in bacteria. Please check this!
4. Figure 1: Replace "DxS" by "Dxs". Different fonts are used, please unify. Gene names should be written in italic fonts. The figure does not show the major metabolic pathways associated with PN synthesis in *E. coli*. Please correct!
5. Statistics: It is unclear how often some experiments have been performed. Please indicate this in all figures where experimental data are shown (e.g., Figure 3d).
6. Page 21 (Page 221), from line 331 on there is a problem with the line numbering.
7. General comment. It would be extremely helpful to look at the cellular amounts of the overproduced proteins by quantitative mass spectrometry. This is nowadays easy to perform and might uncover the potential bottlenecks in the DXP-dependent vitamin B6 pathway.
8. Discussion. Page 32, lines 587 – 589: Why should feedback inhibition of PdxH by PLP affect/reduce the synthesis of PN(P) by the DXP-dependent pathway? This does not make sense.
9. Discussion. The discussion is too long and rather repetitive. What would be the next important steps to solve the issue of low productivity of the engineered strain?
10. The authors did not mention anything about the genetic stability of the engineering best-producing strain. How does the strain behave after serial passaging on rich medium? Did the authors observe colony heterogeneity? This must be tested.

Minor points

1. Pages 3 & 4., lines 64, 67: Replace "gram" by "Gram"
2. Page 4, line 69: delete the extra space: "harbor a"
3. Page 6, line 102: Once 4-hydroxy-threonine is abbreviated 4HT can be mentioned throughout the manuscript. The same is true for 4-hydroxy-threonine or 4-phospho-hydroxy-threonine. Please check and unify!
4. Page 9, Figure 2: Please include the genotypes of the strains. This makes it easier to understand. The genotypes should also be mentioned in the legend to figure 2.
5. Page 10, line 180 and throughout the manuscript: Once "wild type" has been abbreviated, it can be designated as "WT". Please check the manuscript and correct!
6. Page 23, line 416: Replace "DXS" by "Dxs".
7. Page 23, line 426: Please indicate the correct cofactor formula.
8. Page 24, line 437: 5-, 10-, 20-mM PN. Are the lines correct?
9. Page 23, line 460: Can E. coli take up 2-oxoglutarate? Please provide a reference.
10. Figure 5d: I assume that the mutation should be written as "delta-serA" and not "serA-delta".
11. Page 30, line 538: Remove the extra space: "453. 78 mg/L".
12. Page 32, line 576: Replace "pyridoxine" by PN. Check it throughout the manuscript (e.g., page 35, line 639).

Reviewer #2:

Remarks to the Author:

This paper reports the development of a fermentation production method for pyridoxine in *Escherichia coli*. Presumably, all vitamin B6 currently available on the market is chemically synthesized, and it is important to develop a method for supplying it through green chemistry. As for PN, however, it is an inexpensive compound, and it is unclear whether the development of a fermentation production method is worthwhile from the standpoint of production costs.

In *E. coli*, vitamin B6 is synthesized by a multi-step reaction via the DXP-dependent pathway and its concentration is low and tightly regulated. Enzymes on this pathway exhibit low activity and are subject to complex regulation such as product inhibition. The authors searched and expressed the enzymes on the DXP-dependent pathway of various organisms and introduced mutations into these enzymes. They attempted to enhance vitamin B6 production by controlling the expression levels of these enzymes. A number of strains were constructed and evaluated, and those that showed good performance were selected sequentially. Finally, a PN production of 1.4 g/L was achieved, which is probably the highest PN production by microbial production.

I think results are excellent, yet the basis for those results is largely unexplored. In this study, several mutations were introduced into the target protein, which was selected by computational simulations. The effect of each trans/mutated enzyme's expression on the final PN production was evaluated. The authors concluded that the factors contributing to the results were, in most cases, due to increased activity caused by the mutation. In some cases, they stated that the reason was increased protein stability. However, throughout the study, the effects of mutations on the activity, stability, and/or expression levels of the enzyme were not determined experimentally. The authors suggested that their strategy may be applicable to other compounds, however, without direct data to demonstrate the

reliability of the simulations, the usefulness of this methodology may not be shown.

Construction of parallel metabolic pathway

Line170: The pdxH deficient strain supposedly is unable to synthesize PLP from PN, but why is growth partially restored by the addition of high concentrations of PN?

Line171-172: Which data supports this conclusion?

Line175: Please describe why you selected pdxS-T from Solanum.

Line188: The authors overlooked the presence of PdxI. The E. coli K strain possesses the PL reductase PdxI. Thus, the PLP synthesized by pdxST is likely converted to PL→PN, and the observed increase in PN production is probably not coming from the DXP-dependent pathway, but from pdxST reaction. The presence of pdxI could reasonably explain why PN production is increased in the strains expressing pdxST. This comment can be checked by observing whether the pdxI-deficient strain also shows an increased PN production.

Line196: What is the reason for choosing the pta locus?

Effects of overexpression of pdxA, pdxJ, and their mutants

Line233: It is described that the codon-optimized pdxA and pdxJ genes were cloned into pRSFDuet-1, which was then introduced into strain LL006 to increase PN productivity. This plasmid is probably a T7 promoter-controlled expression vector; target genes should be barely expressed in strain MG1655, so please describe why the authors used this plasmid and how expression is induced and controlled.

There appears to be no correlation between the estimated binding energy of the various mutants and the PN titer, but this estimate is used to interpret the results. The expression levels of the enzymes, and the enzyme activity of the mutant enzymes, have not been measured. I think that measurement of expression levels and of the activity of pdxA and pdxJ is needed to support the authors' conclusions.

Overexpression of epd and dxs

Although it has been concluded that the introduction of epd from Glaciecoda increased PN titer because of their increased affinity for NAD, no direct data have been presented to support this. Expression levels and enzyme's activity should be measured. How high are the predicted concentrations of E4p and NAD in this bacterium? Are these values consistent with the data obtained?

PdxB and SerC

Line435&fig5a: It does not appear to me that PN production is increased by α -KG, is this a statistically significant difference? Is it really possible that α -KG added to the medium can lead to an increase in intracellular α -KG levels? Is there sufficient flux in the Ser synthesis pathway to impact α -KG levels through overexpression of SerC?

SerC accepts α -KG as a substrate. Therefore, an increase in the intracellular concentration of α -KG might not favor PN synthesis in response to SerC reaction.

Line464: The initial growth rate is unchanged in the serA-deficient strain compared to the cont., but the final OD value seems to be lower. I do not think Ser is added to the medium. Is this caused by Ser toxicity?

Line472-474: Please explain the link between the growth defect of serA and the increase in PN productivity by overexpressing serC.

Multimodule optimization

Regarding Fig. 6, please state clearly and precisely which strain is used for each data.

Fig6b: The "con." is the data with strain LL372. Which strains were used for the data shown below? with strains LL373~LL377? In supplementary information, it is shown that they are originating from strain LL370. The stains seem to have two plasmids with identical backbones, and does not harbor pdxA-pdxJ expression plasmid. Or are they derived from LL372? If the author was correct, how were they produced?

Fig6d: Is the genotype of Con. (LL377) correct? If the genotype was "LL372, harboring p15ASI-Ptac-epd (Gni)-pdxB(Eco) -dxs(Sme)-PJ231119 serC (Eco)", why the PN titer was significantly different from the data of Fig 6b?

It is interesting that productivity is enhanced by the expression of enzymes derived from other organisms or mutant enzymes, rather than by simply controlling the expression level of E. coli enzymes. Throughout this study, it is not clear why the expression of mutant enzymes or enzymes from other organisms resulted in this outcome.

Lines 532-533,542-543: Here the purpose of mutant enzyme expression is described as to improve expression performance, but in the previous part the purpose was to increase enzyme activity. If so, what is the advantage of using this mutant compared to the wild type enzyme? As a control, I think LL379 should be included.

Reviewer #3:

Remarks to the Author:

Key Results:

The approach is clearly explained and documented with mixed approaches of step by step improvement with single enzymes improvement (protein engineering or heterologous enzyme) and combinatorial expression. Protein improvement strategies are based either on protein engineering based on in silico simulation or heterologous enzyme replacement. All steps seems rationally designed to reduce the number of iteration or libraries sizes. Highly appreciated the step by step explanation for protein engineering simulation. The decoupling of the two vitamin B6 pathways in order to keep low level of PLP for growth maintenance and push the PN pathway for production is of high interest.

Validity & analytical approach:

All data presented in graphs show the results of independent biological replicate and corresponding standard deviation are mentions. The OD measured sometimes appear higher or lower than in the majority of the other experiments: significantly lower on the supplementary Fig. 7 and higher on the Fig. 5d; and slightly higher on the Fig. 5a and 6. Not compromising the results validity to my point of view but: (i) could be nice to add the details about the scale used (24 deep well plates or flask scale) that can explain the OD differences, (ii) comment on the OD variation on a same scale can be of high interest since growth of the production strain is central on the strategy describe on the paper.

Significance:

Decoupling growth to production is commonly employed to improve fermentation production and even more in the case of toxic final product or intermediates. As far as I know, the more often the decoupling strategy are based on carbon sources switch, metabolic shift or induction process. Here, the decoupling growth based on two parallel pathways and modulated differently appear innovative.

Data and methodology:

Strategy and rational are clearly explained and documented. The step by step explanations are easy to assess. Supplementary data provided are essential for a complete view of the approach and the results. The protein engineering simulation take lot of importance in the paper and is clearly explained, nevertheless I have no strong expertise is that simulation field and cannot tell more about the relevance of the methods chosen. I have one comment on the strategy employed to improve the different enzymes: some have been chosen to be improved using modeling and protein engineering and others using heterologous gene replacement. Why one or the other strategy has been chosen is not justified.

Also attached a word document with additional specific comments that to my point of view can be improved (to complete the results or to go further in the conclusions).

Paper review / additional comments and suggested improvements

Protein engineering and iterative multimodule optimization for vitamin B6 production in Escherichia coli

Linxia Liu, Jinlong Li, Yuanming Gai, Zhizhong Tian, Yanyan Wang, Tenghe Wang, Qianqian Yuan, Hongwu Ma, Sang Yup Lee & Dawei Zhang

- Line 169-170 + fig2 b

Why supplemented the mutant strain with PN and not PL. Referring to the reference 35, they show that PL can restore phenotype in LB with Cysteine. Have you taking a look on the morphology of the mutant *E.coli* (described as smaller colonies in ref 35), that could explain the lower OD observed. Can you comment on this and why PN feeding can lead to increase OD (other pdx family enzyme?).

- Line 185-186 + fig2 b

It is hypothesized here that PLP produced by the heterologous expression of pdxST from *B.subtilis* is able to fully restore growth of the pdxH mutant. Have you some data to share about PLP produced in that strain? Have you an assay showing the similar growth phenotype can be obtained with the pdxH mutant supplemented with PL (or PLP) or complemented with the pdxST heterologous expression?

- Line 188-189 + supplementary Fig1

Could be interested to normalize PN production with biomass and comment on this. Since the two pathways has been separated by the pdxH KO, PN increase should be linked to a better growth, if it's still enhanced in that case can you comment and hypothesized about the flux in that double pathway context.

- Line 195-196

Why pta locus has been targeted for pdxP insertion? Hypothesis and comment can be appreciated and even more if relevant for fermentation process.

- Line 283-285 + fig 3a

Mutants G207A and V35I are not shown on the figure 3a. On the figure 3a, probably for an esthetic concern, the OD scale starts with negative values, found it more relevant to start at zero instead.

- Paragraph on Epd Line 361-401

Heterologous Epd are selected regarding evolutionary divergence between sequences and better structural stability. It is hypothesized, regarding binding energies of NAD⁺ and E4P, that the increased PN titer is mainly due to NAD⁺ binding energy lower in *G.nitratireducens* than *E.coli*. Is there data about specific binding energies with NAD⁺ and E4P for the others heterologous Epd tested? Particularly, can others improved PN titer Epd candidates can reinforce the hypothesis of the increased stable binding of NAD⁺ is responsible for the enhanced activity? Further energy binding analysis and comment around the results of the panel can lead to interesting discussion.

- Line 407-410

What are the criteria applied to select the Dxs heterologous enzymes?

- Line 468-470

The results showed that growth was slightly better than with serine addition, but ~~serine-glycine~~ (?) addition was also unable to completely restore growth (Fig. 5d).

- Paragraph on SerA / SerC strategy Line 453-474

Since SerC is described to be a promiscuous enzyme with several defined substrates and different fluxes with the lower one for the vitamin B6 pathway. Why don't try a protein engineering strategy to increase substrate specificity for the vitamin B6 pathway and perhaps express two SerC enzymes, one for global metabolism and to maintain a good growth rate and the other one to improve flux in the vitamin B6 pathway? Could be interesting to discuss on that.

- Fig 7

The growth curve looks atypical with an intermediate low OD stationary phase between 12 and 36h and then a restart of growth. Can you comment on this? Is there dissolved oxygen data and PLP analysis to share that can feed the discussion and hypothesis around the fermentation results? Since PLP and PN synthesis has been disconnected in order to separate growth and PN synthesis, could be interesting to check this hypothesis with protein expression level and make some hypothesis regarding growth behavior at larger scale. From such discussion, further improvement can be discussed to improve even more at larger scale.

- Line 594-598 + supplementary fig 9

Results shown on this figure are of high interest and further discussion and hypothesis can be done. Results show that overexpression of PdxP from *E. meliloti* leads to increase PN accumulation (LL005 with the two plasmids vs LL006 with the two plasmids); the results also shows that in absence of pdxP from *E. meliloti*, *E. coli* is able to produce PN and even more with pdxH KO (WT with the two plasmids vs LL005 with the two plasmids). Can you comment and make hypothesis on this? Is there any PNP and PLP measurement available to feed the comment?

- Why the carbon source for the fed batch fermentation is glycerol? Can you justify this choice.

Response to reviewers for “Protein engineering and iterative multimodule optimization for vitamin B₆ production in *Escherichia coli*”

Reviewer #1 (Remarks to the Author):

With great interest I was reading the manuscript by Liu and co-worker about the engineering attempts of *Escherichia coli* for overproducing the commercially attractive B₆ vitamer pyridoxine (PN). For this purpose, various genetic modules were constructed to overexpress the genes of the DXP-dependent vitamin B₆ pathway in *E. coli*. Moreover, the *pdxST* operon from *Bacillus subtilis* was co-expressed in the background of a *pdxH* deletion mutant. By following this approach, the authors managed to produce 1.4 g/L of PN, which is the highest reported vitamin B₆ titer. The amount of work to enhance the production of PN is quite large. However, I have several issues that need to be addressed by the authors.

Major points 1. Previously, it has been shown that the *pdxST* genes from *B. subtilis* can be functionally expressed in *E. coli* for producing the biologically active B₆ vitamer PLP. Therefore, it is unclear why the authors tested the *pdxST* genes from different organisms. Some of the data presented in Figure 2 can be removed.

Response: Thank you for the comment. The goal of testing the *pdxST* genes from different organisms was to find a better combination of *pdxST* operons. Potato (*Solanum tuberosum* L.) is one of the major food crops used worldwide and can provide an excellent dietary source of vitamins including vitamin B₆. Therefore, we selected the genes from *S. tuberosum* to test the production of PLP. However, it failed to be expressed well in *E. coli* compared to the *pdxST* genes from *B. subtilis*. We have removed the related data in the revised manuscript.

2. In Figure 2b it is shown that the B₆ vitamer PN complements PLP auxotrophy of the *E. coli* *pdxH* mutant. How is this possible? The *pdxH* mutant cannot convert PNP to the biologically active B₆ vitamer PLP. How much PN was added to the medium? Usually,

commercially available PN contains a 1000-fold lower amount of PL. The *pdxH* mutant can easily be complemented with PL.

Response: Thank you for pointing this out. The reviewer is correct, and we have made the correction accordingly in the revised manuscript. We used a high concentration of PN up to 100 μM . When the concentrations of PL and PN were reduced to 1 μM , the growth defects can indeed be compensated by PL, but not by PN (Fig. R1a). The addition of 100 μM PN to the *ΔpdxH* mutant of *E. coli* can also compensate for PLP defects, which may be due to the low amount of PL in commercial PN rather than the role of PN itself, as noted by the reviewer. We have updated Fig.2b in the revised manuscript.

Fig. R1a. The cell growth of the *ΔpdxH* mutants (LL001) with or without *pdxST* expression. LL003, *ΔpdxH::pdxST-1*; LL005, *ΔpdxH::pdxST-2*. PL or PN was added to the medium at a concentration of 1 or 100 μM .

3. Pages 3-4, lines 44-70: The introduction or at least the red line of the introductory text is relatively close to existing reviews about vitamin B₆ metabolism in bacteria. Please check this!

Response: Thank you. We have revised the text to address your concerns and hope that it is now clear. Please see pages 3-4 of the revised manuscript, lines 49-68.

4. Figure 1: Replace “DxS” by “Dxs”. Different fonts are used, please unify. Gene names should be written in italic fonts. The figure does not show the major metabolic pathways associated with PN synthesis in *E. coli*. Please correct!

Response: Thank you. We have corrected the “DxS” into “Dxs” and checked the others throughout the manuscript.

The Fig. 1 has been updated in the revised manuscript.

Revised Fig. 1. Major metabolic pathways associated with PN biosynthesis in *E. coli*, metabolic engineering approaches applied to overproduce PN, and carbon flux from glycerol toward PNP starting from glycerol.

5. Statistics: It is unclear how often some experiments have been performed. Please indicate this in all figures where experimental data are shown (e.g., Figure 3d).

Response: Thank you for the comment. Experiments were performed in duplicates or triplicates. Error bars indicate s.d.

Fig. 3d displayed the PN titer distribution of four rounds PdxJ mutants including a total of 212 mutants: 26 single mutants (round 1), 62 double mutants (round 2),

60 triple mutants (round 3), and 64 quadruple mutants (round 4). All the detailed data are listed in Supplementary Data 2.

6. Page 21 (Page 221), from line 331 on there is a problem with the line numbering.

Response: We corrected the line numbers.

7. General comment. It would be extremely helpful to look at the cellular amounts of the overproduced proteins by quantitative mass spectrometry. This is nowadays easy to perform and might uncover the potential bottlenecks in the DXP-dependent vitamin B₆ pathway.

Response: Thank you for the comment. Proteomic samples from the fermentation of LL006 with empty vectors or high-producing vectors (LL388) at 16 h were prepared and analyzed for quantitative proteomic analysis. The overproduced proteins were evaluated based on iBAQ (intensity based absolute quantification) values (Table R1). Please see page 38 of the revised manuscript, lines 711–721.

Table R1. The cellular amounts of the overproduced proteins (Mol proportion).

Protein names	LL006 with empty vectors	LL388
PdxA2	0.002%	0.945%
PdxJ1	0.014%	3.706%
Gni_Epd	ND ¹	0.925%
Eme_Dxs	ND ¹	0.156%
PdxB	0.018%	1.021%
SerC	0.036%	0.322%

¹ND, Not detected.

Based on the cellular protein quantification results, it was evident that the Dxs and SerC proteins could potentially be the bottlenecks in the DXP-dependent vitamin B₆ pathway due to their lower expression compared to other proteins. Consequently, the next crucial step is to elevate the expression levels of Dxs and SerC in order to enhance PN production even further.

8. Discussion. Page 32, lines 587–589: Why should feedback inhibition of PdxH by PLP affect/reduce the synthesis of PN(P) by the DXP-dependent pathway? This does not make sense.

Response: Thank you for pointing this out. The reviewer is correct, and we have made the correction accordingly in the revised manuscript (lines 667-669 page 36).

“In our study, the synthesis of PLP required for cell growth and PN production were decoupled and modulated to maintain cell growth and enhance PN production using two parallel biosynthetic pathways.”

9. Discussion. The discussion is too long and rather repetitive. What would be the next important steps to solve the issue of low productivity of the engineered strain?

Response: We apologize for the lack of sufficient information in the Discussion section of our previous manuscript. In the revised version, we have made corrections to ensure a more relevant and streamlined discussion. To tackle the problem of low production efficiency in engineered bacteria, several crucial steps need to be taken. These include the modulation or upregulation of specific protein expressions, cofactor engineering, construction of a plasmid-free strain, and the scale-up of the fermentation process. Among these steps, the optimization of the fermentation process is particularly challenging, and we are currently working diligently to improve the situation (lines 711-724 pages 38-39).

10. The authors did not mention anything about the genetic stability of the engineering best-producing strain. How does the strain behave after serial passaging on rich medium? Did the authors observe colony heterogeneity? This must be tested.

Response: Thank you for the comment. We conducted a series of continuous passages, spanning 1 to 100 generations, on FM1.4 fermentation medium, a nutrient-rich medium. During the 40th to 100th generations, we observed a significant decrease of PN yield, and approximately 15% of the cells lost the plasmids (Supplementary Fig. 9 in the revised Supplementary information). To examine the colony morphology changes, we utilized Leica stereo microscopes.

The results revealed that a fraction of colonies (~15%) exhibited larger size and thinner edges, suggesting the presence of mutants that had lost the plasmids. However, there were no significant differences observed in the growth curves among the various generations, although the 100th and 60th generation cells exhibited slightly faster growth compared to the parental strain (1st generation). These are added in the revised manuscript (lines 617-632 pages 33-34).

Supplementary Fig. 9. Genetic stability of the engineered best-producing strain. a and b The cell growth (OD₆₀₀) and PN titer of the evolved strains of LL388, which were sequentially transferred in FM1.4 medium. **c** The colony heterogeneities of the parental strain and the evolved mutants. **d** The growth curves of the cells of the 1st, 60th and 100th generations. The 60th and 100th generations were about the fifth- and eighth-day passaging, respectively.

Minor points:

1. Pages 3 & 4., lines 64, 67: Replace “gram” by “Gram”

Response: Corrected.

2. Page 4, line 69: delete the extra space: “harbor a”

Response: Corrected.

3. Page 6, line 102: Once 4-hydroxy-threonine is abbreviated 4HT can be mentioned throughout the manuscript. The same is true for 4-hydroxy-threonine or 4-phospho-hydroxy-threonine. Please check and unify!

Response: We would like to thank the reviewer for carefully reviewing our manuscript. We have made changes in the revised manuscript.

4. Page 9, Figure 2: Please include the genotypes of the strains. This makes it easier to understand. The genotypes should also be mentioned in the legend to figure 2.

Response: Done as suggested.

5. Page 10, line 180 and throughout the manuscript: Once “wild type” has been abbreviated, it can be designated as “WT”. Please check the manuscript and correct!

Response: Corrected.

6. Page 23, line 416: Replace “DXS” by “Dxs”.

Response: Corrected.

7. Page 23, line 426: Please indicate the correct cofactor formula.

Response: Corrected.

8. Page 24, line 437: 5-, 10-, 20-mM PN. Are the lines correct?

Response: Corrected.

9. Page 23, line 460: Can *E. coli* take up 2-oxoglutarate? Please provide a reference.

Response: *E. coli* can take up 2-oxoglutarate (2-OG). Reference is provided.

1. Zhang, C., Wei, Z.-H. & Ye, B.-C. Quantitative monitoring of 2-oxoglutarate in *Escherichia coli* cells by a fluorescence resonance energy transfer-based biosensor. *Appl. Microbiol. Biotechnol.* 97, 8307-8316 (2013).

“Exogenous 2OG addition induced a decrease in the YFP/CFP emission ratio (i.e., an increase in the intracellular 2OG level), which indicates the transport of 2OG across the membrane”. -From the above reference.

We also monitored changes in 2-OG levels in living *E. coli* cells after the addition of 2-OG by HPLC. When 50 mM 2-OG was added, the 2-OG concentration in the culture medium decreased after 48 h of cultivation, which suggested that *E. coli* can take up 2-OG (Fig. R2).

Fig. R2. The 2-OG concentration in the culture medium at 0 h (before cultivation) and 48 h.

10. Figure 5d: I assume that the mutation should be written as “delta-serA” and not “serA-delta”.

Response: Corrected.

11. Page 30, line 538: Remove the extra space: “453. 78 mg/L”.

Response: Corrected.

12. Page 32, line 576: Replace “pyridoxine” by PN. Check it throughout the manuscript (e.g., page 35, line 639).

Response: Corrected.

Reviewer #2 (Remarks to the Author):

This paper reports the development of a fermentation production method for pyridoxine in *Escherichia coli*. Presumably, all vitamin B₆ currently available on the market is chemically synthesized, and it is important to develop a method for supplying it through green chemistry. As for PN, however, it is an inexpensive compound, and it is unclear whether the development of a fermentation production method is worthwhile from the standpoint of production costs.

Response: Thank you for bringing this to our attention. As mentioned by the reviewer, PN is an inexpensive compound, with a cost of approximately \$23 per kilogram according to China Customs data in 2021. However, the global market for vitamin B₆ was valued at \$81 million in 2020 and is projected to reach \$138.4 million by the end of 2027, with a compound annual growth rate (CAGR) of 8.8% during the period 2021-2027, as reported by MarketWatch (source: <https://www.marketwatch.com/press-release/global-vitamin-b6-market-2023-analysis-of-latest-industry-growth-and-share-by-2031-2023-06-03>).

It is worth noting that vitamin B₆ is currently produced through complete chemical synthesis, utilizing the Diels–Alder reaction with 4-methyl-5-cyanooxazole. The chemical process used too many solvents such as benzene and toluene, which belong to Class 1 and 2 solvents labeled in the United States Pharmacopeia (USP), respectively. The industrial by-products were difficult to treat and costly, such as large amounts of phosphate. Additionally, high-temperature and chlorination reactions increased the safety risk¹. Given these factors, there is a pressing need for an alternative fermentation-based production method for vitamin B₆, particularly for chemical synthesis production companies such as major manufacturers like Tianxin Pharmaceutical, DSM Nutritional Products Ltd., and Guangji Pharmaceutical. Interestingly, DSM had already ventured into fermentation production of vitamin B₆ over 20 years ago, as evidenced by publicly available patents and published articles (references 2-6).

However, the yield achieved at that time did not meet the commercial expectations, which emphasizes the importance of surpassing the productivity of chemical

synthesis. To be commercially viable, production levels exceeding 10 g/L within 48 hours are required (reference 5). In our research, we are committed to maximizing the yield, and the findings presented in our manuscript serve as a strong foundation for the environmentally friendly bio-based production of vitamin B₆ and its future industrialization. We have included this relevant information in the introduction section of the revised manuscript (lines 61-68 pages 3-4).

1. Eggersdorfer, M. et al. One hundred years of vitamins—a success story of the natural sciences. *Angew. Chem. Int. Ed.* 51, 12960-12990 (2012).
2. Hoshino, T., Ichikawa, K., Nagahashi, Y. & Tazoe, M. Microorganism and process for preparing vitamin B₆. US Patent Application US2006/0127992 A1. (2006a).
3. Hoshino, T., Ichikawa, K. & Tazoe, M. Recombinant microorganism for the production of vitamin B₆. US Patent Application US2006/0228785 A1. (2006b).
4. Hoshino, T., Nagahashi, Y. & Tazoe, M. Gene encoding vitamin B₆ phosphate phosphatase and use thereof. US Patent Application US7598066 B2. (2009).
5. Commichau, F. M. et al. Overexpression of a non-native deoxyxylulose-dependent vitamin B₆ pathway in *Bacillus subtilis* for the production of pyridoxine. *Metab. Eng.* 25, 38-49 (2014).
6. Commichau, F. M. et al. Engineering *Bacillus subtilis* for the conversion of the antimetabolite 4-hydroxy-l-threonine to pyridoxine. *Metab. Eng.* 29, 196-207 (2015).

In *E. coli*, vitamin B₆ is synthesized by a multi-step reaction via the DXP-dependent pathway and its concentration is low and tightly regulated. Enzymes on this pathway exhibit low activity and are subject to complex regulation such as product inhibition. The authors searched and expressed the enzymes on the DXP-dependent pathway of various organisms and introduced mutations into these enzymes. They attempted to enhance vitamin B₆ production by controlling the expression levels of these enzymes. A number of strains were constructed and evaluated, and those that showed good performance were selected sequentially. Finally, a PN production of 1.4 g/L was achieved, which is probably the highest PN production by microbial production.

I think results are excellent, yet the basis for those results is largely unexplored. In this

study, several mutations were introduced into the target protein, which was selected by computational simulations. The effect of each trans/mutated enzyme's expression on the final PN production was evaluated. The authors concluded that the factors contributing to the results were, in most cases, due to increased activity caused by the mutation. In some cases, they stated that the reason was increased protein stability. However, throughout the study, the effects of mutations on the activity, stability, and/or expression levels of the enzyme were not determined experimentally. The authors suggested that their strategy may be applicable to other compounds, however, without direct data to demonstrate the reliability of the simulations, the usefulness of this methodology may not be shown.

Response: Thank you for the comment. Firstly, we had constraints in purchasing additional substrates for our study. We were only able to acquire substrates for Epd and measure its activity, as discussed in the later part of this response. Consequently, we were unable to directly demonstrate the higher enzyme activities of the PdxA and PdxJ mutations, which resulted in increased production yields, through kinetics data of the enzymes. However, we made efforts to establish a correlation between production yields and enzyme activities through various experiments. We examined the expression levels of both PdxA and PdxJ, as well as their respective high-yielding mutants. Interestingly, the expression levels decreased significantly in some cases (E214N in *pdxA* and E104T/I218L/G194C and I13V/L212I/E104T in *pdxJ*) (Supplementary Fig. 3 in the revised Supplementary information). These observations indirectly suggest that the enhanced production yields may be attributed to an increase in enzyme catalytic efficiency. We have incorporated these findings and modifications in the revised manuscript (lines 294-297 page 16, lines 312-315 page 17, lines 367-372 pages 19-20, and Supplementary Fig. 3).

On a different note, we must acknowledge that while improving enzyme activity can greatly contribute to production yields, it may not guarantee a continuous increase in those yields. Therefore, our approach focuses on constructing an

intelligent mutation library enriched with beneficial mutations, rather than solely relying on computational screening to identify enzymes with the highest activity. While an enzyme selected through computational screening may exhibit superior activity, its impact within the pathway may not solely depend on enzyme activity alone but also on its compatibility with other enzymes. Thus, it is crucial to have a mutation library with a high enrichment of positive mutations. This library may encompass different mutations, each with high activity, which can be further screened to identify combinations of mutations that are well-suited to the pathway. The method we employed for enzyme design is an effective strategy for increasing the proportion of beneficial mutations within mutation libraries (reference 1). This enables the generation of a greater number of high-yield mutant strains through the synergistic adaptation of different enzymes. We have incorporated these changes in the revised manuscript (lines 217-225 page 12).

In addition, the K_m values of the Epd enzyme with the best binding energy in molecular dynamics (MD) simulations were consistent with the experimental results. Both NAD^+ and E4P affinities were improved, and the improvement was more pronounced for NAD^+ (E4P: Eco_Epd $K_m = 0.969$ mM, Gni_Epd $K_m = 0.737$ mM; NAD^+ : Eco_Epd $K_m = 0.717$ mM, Gni_Epd $K_m = 0.051$ mM). It is important to highlight that despite the reduction in protein expression (Supplementary Fig. 3c), the catalytic capability of Gni_Epd was significantly improved, particularly for the substrate E4P, almost 3-fold enhancement in the k_{cat}/K_m value for E4P (Table 1). This improvement in the catalytic capacity of Gni_Epd, along with enhanced binding and recruitment of E4P and NAD^+ , played a major role in the observed increase in product yield.

Furthermore, we observed that the k_{cat} values obtained in our study significantly differed from those reported in previous studies, whereas the K_m values were similar. Specifically, in our study, the K_m value was measured to be 0.969 mM and the k_{cat} value was 0.15 s^{-1} , whereas in previous studies (references 2 and 3), the reported K_m values ranged from 0.510 to 0.96 mM, and the k_{cat} values ranged from

20 to 200 s⁻¹. The experimental data and measurement methods were thoroughly reviewed and validated to ensure the accuracy and reliability of the data. Based on the newly obtained measurements, it can be inferred that the binding and catalysis of Epd may represent a new bottleneck in the pathway. Thus, further optimization of the k_{cat} value and the binding interactions of E4P and the coenzyme could effectively enhance the substrate flow towards the PN pathway, ultimately leading to improved PN production. All the relevant changes have been indicated in red in the revised manuscript (lines 432-452 pages 23-24, Supplementary Fig. 3c, and Table 1).

Supplementary Fig 3. a The expression of PdxA and its mutants. b The expression of PdxJ and its mutants. c The expression of Eco_Epd and Gni_Epd. The target protein is marked with a red arrow. PdxA con. and PdxJ con. represent the native proteins.

Table 1. Kinetic parameters of purified Eco_Epd and Gni_Epd toward E4P and NAD⁺.

Epd	E4P			NAD ⁺		
	K_m (mM)	k_{cat} (s ⁻¹)	k_{cat}/K_m (s ⁻¹ /mM)	K_m (mM)	k_{cat} (s ⁻¹)	k_{cat}/K_m (s ⁻¹ /mM)
Eco_Epd	0.969	0.150	0.155	0.717	3.232	4.511
Gni_Epd	0.737	0.323	0.438	0.051	0.313	6.137

1. Li, J. et al. Going beyond the local catalytic activity space of chitinase using a simulation-based iterative saturation mutagenesis strategy. *ACS Catal.* 12, 10235-10244 (2022).
2. Zhao, G., Pease, A. J., Bharani, N. & Winkler, M. E. Biochemical characterization of *gapB*-encoded erythrose 4-phosphate dehydrogenase of *Escherichia coli* K-12 and its possible role in pyridoxal 5'-phosphate biosynthesis. *J. Bacteriol.* 177, 2804-2812 (1995)
3. Boschi-Muller, S., Azza, S., Pollastro, D., Corbier, C. & Branlant, G. Comparative enzymatic properties of GapB-encoded erythrose-4-phosphate dehydrogenase of *Escherichia coli* and phosphorylating glyceraldehyde-3-phosphate dehydrogenase. *J. Biol. Chem.* 272, 15106-15112 (1997).

Construction of parallel metabolic pathway

Line170: The *pdxH* deficient strain supposedly is unable to synthesize PLP from PN, but why is growth partially restored by the addition of high concentrations of PN?

Response: Thank you for pointing this out. The reviewer is correct, and we have made the correction accordingly in the revised manuscript. We used a high concentration of PN up to 100 μM . When the concentrations of PL and PN were reduced to 1 μM , the growth defects can indeed be compensated by PL, but not by PN (Fig. R1a). The addition of 100 μM PN to the Δ *pdxH* mutant of *E. coli* can also compensate for PLP defects, which may be due to the low amount of PL in commercial PN rather than the role of PN itself, as noted by the Reviewer #1. We have updated Fig.2b in the revised manuscript.

Fig. R1a. The cell growth of the *ΔpdxH* mutants (LL001) with or without *pdxST* expression. LL003, *ΔpdxH::pdxST-1*; LL005, *ΔpdxH::pdxST-2*. PL or PN was added to the medium at a concentration of 1 or 100 μM.

Line171-172: Which data supports this conclusion?

Response: Thank you for the comment. The conclusion mentioned was derived from another reference, rather than based on our own experimental findings. Thus, we now deleted this sentence in the revised manuscript.

Line175: Please describe why you selected *pdxS-T* from *Solanum*.

Response: Thank you. Potato (*Solanum tuberosum* L.) is one of the major food crops worldwide. A billion of people depend on the potato for daily food for its excellent dietary source of vitamins. Therefore, we selected the *pdxST* genes from *S. tuberosum* to test the production of PLP. However, it was difficult to meet the needs of PLP in *E. coli* even after optimization compared to the *pdxST* genes from *B. subtilis*. According to the comment from Reviewer #1, we deleted the related data in the revised manuscript.

Line188: The authors overlooked the presence of PdxI. The *E. coli* K strain possesses the PL reductase PdxI. Thus, the PLP synthesized by *pdxST* is likely converted to PL→PN, and the observed increase in PN production is probably not coming from the DXP-dependent pathway, but from *pdxST* reaction. The presence of *pdxI* could reasonably explain why PN production is increased in the strains expressing *pdxST*. This comment can be checked by observing whether the *pdxI*-deficient strain also shows an increased PN production.

Response: Thank you for pointing this out. The PdxST enzyme complex functions at a relatively low rate, and its expression was introduced only to fulfill the cellular requirement of PLP in our initial design. We attempted to delete or overexpress the *pdxI* gene, but it had minimal impact on PN production (Fig. R3).

Fig. R3. The PN titer and cell growth obtained with the $\Delta pdxI$ and $pdxI^{OE}$ mutants. The overexpression of $pdxI$ led to a decrease in PN production, possibly due to the metabolic burden imposed by the use of the high-copy pRSFDuet-1 plasmid.

Line196: What is the reason for choosing the *pta* locus?

Response: Thank you for the comment. *E. coli* produces acetate, which is undesirable as it inhibits growth and protein synthesis in aerobic cultures. In the field of metabolic engineering, it is common to delete genes involved in key central metabolites, such as pyruvate and acetyl-CoA, to improve desired metabolic pathways. Therefore, we chose to insert Eme *pdxp* into the *pta* locus in our study. We have made revisions to the text (lines 209-211 page 11) to provide clarity on this matter. Notably, the disruption of the *pta* gene had no effect on the fermentation process for PN, as indicated in Supplementary Fig. 1c. Several studies have demonstrated the selection of the *pta* gene for metabolic engineering purposes to enhance the production of high-value compounds as shown below.

1. Singh, A., Cher Soh, K., Hatzimanikatis, V. & Gill, R. T. Manipulating redox and ATP balancing for improved production of succinate in *E. coli*. *Metab. Eng.* 13, 76-81 (2011).
2. Yang, T., Wu, P., Zhang, Y., Cao, M. & Yuan, J. High-titer production of aromatic amines in metabolically engineered *Escherichia coli*. *J. Appl. Microbiol.* 33, 2931-2940 (2022).
3. Zhou, J. et al. An acetate-independent pathway for isopropanol production via HMG-CoA in *Escherichia coli*. *J. Biotechnol.* 359, 29-34 (2022).

4. Vikromvarasiri, N., Shirai, T. & Kondo, A. Metabolic engineering design to enhance (R, R)-2,3-butanediol production from glycerol in *Bacillus subtilis* based on flux balance analysis. *Microb. Cell Fact.* 20, 196 (2021).

Supplementary Fig. 1c. The PN titer and cell growth of LL005 and LL005b (LL005 *Apta*).

Effects of overexpression of *pdxA*, *pdxJ*, and their mutants

Line233: It is described that the codon-optimized *pdxA* and *pdxJ* genes were cloned into pRSFDuet-1, which was then introduced into strain LL006 to increase PN productivity. This plasmid is probably a T7 promoter-controlled expression vector; target genes should be barely expressed in strain MG1655, so please describe why the authors used this plasmid and how expression is induced and controlled.

Response: Sorry for the unclear description in the original submission. pRSFDuet-1 is a commonly used expression vector for producing recombinant proteins in *E. coli*. In our study, high-copy-number plasmid pRSFDuet-1 was used to overexpress the genes encoding PdxA and PdxJ to achieve high metabolic fluxes. However, the inducible T7 promoter together with the *lac* operator, and ribosome binding site sequences were replaced by the constitutively expressed promoter J23119 (lately optimized, Fig. 6c) and the synthetic ribosome binding site sequences. We have revised the text clearly in the manuscript accordingly (lines 802-805 page 42).

There appears to be no correlation between the estimated binding energy of the various mutants and the PN titer, but this estimate is used to interpret the results. The expression

levels of the enzymes, and the enzyme activity of the mutant enzymes, have not been measured. I think that measurement of expression levels and of the activity of *pdxA* and *pdxJ* is needed to support the authors' conclusions.

Response: Thank you for the comment. As explained earlier, we were unable to purchase the substrates of PdxA and PdxJ. So, we could not directly demonstrate whether the mutants that increased production had higher enzyme activities based on enzyme kinetics data. However, we still made efforts to demonstrate the relationship between production and enzyme activity through additional experiments. We examined the expression levels of both PdxA and PdxJ, as well as their respective high-yielding mutants. Interestingly, the expression levels decreased significantly in some cases (E214N in *pdxA* and E104T/I218L/G194C and I13V/L212I/E104T in *pdxJ*) (Supplementary Fig. 3 in the revised Supplementary information). These observations indirectly suggest that the enhanced production yields may be attributed to an increase in enzyme catalytic efficiency. We have incorporated these findings and modifications in the revised manuscript (lines 294-297 page 16, lines 312-315 page 17, lines 367-372 pages 19-20, and Supplementary Fig. 3a).

On a different note, during the initial design phase, we employed Rosetta's binding energy score as a preliminary evaluation metric. Although we acknowledge the limitations in using binding energy scores as an exact correlate of enzyme activity, we considered that Rosetta's binding energy score is a valuable tool for assessing the binding energy relationships between small molecules and proteins, particularly in cases where protein structures have limited conformational sampling. Thus, it served as a classification criterion to exclude unfavorable mutations in our study. This approach aligns with our previous method (reference 1), which may be influenced by the degree of sampling. However, due to computational constraints, performing detailed sampling for each mutant was not feasible, considering the large number of mutants involved. Therefore, in the initial design stage, we utilized Rosetta's binding energy score as an initial classification criterion, considering that mutants with better activity than the wild

type in terms of binding energy score may exhibit improved activity to some extent. This allowed us to generate a library of mutants with potentially enhanced activity, which were further verified and screened through experimental validation. Furthermore, we utilized molecular dynamics (MD) simulation methods for a more comprehensive sampling of protein structures. The binding energy evaluation of high-yield mutants using MD methods better reflects the relationship between activity and binding energy compared to Rosetta's static scoring method. Therefore, during the analysis of high-yield mutants, we employed MD methods to further evaluate their activity. The results obtained from this evaluation were consistent with the experimental production, suggesting that the mutants indeed exhibited improved activity to some extent, leading to an increase in production. It is important to clarify that the most active mutant is not necessarily the one that yields the highest production. Moreover, a library with a high enrichment of positive mutations is better suited for screening high-yield mutant strains.

1. Li, J. et al. Going beyond the local catalytic activity space of chitinase using a simulation-based iterative saturation mutagenesis strategy. *ACS Catal.* 12, 10235-10244 (2022).

Overexpression of *epd* and *dxs*

Although it has been concluded that the introduction of *epd* from *Glaciecola* increased PN titer because of their increased affinity for NAD, no direct data have been presented to support this. Expression levels and enzyme's activity should be measured. How high are the predicted concentrations of E4p and NAD in this bacterium? Are these values consistent with the data obtained?

Response: Thank you very much for your valuable suggestions. We have done kinetic experiments on Epd from *E. coli* (Eco) and *Glaciecola nitratireducens* (Gni) (Table 1). From the experimental data, it can be seen that Gni_Epd improved the binding ability to both E4P and NAD⁺ to some extent, and showed better binding ability to NAD⁺ (E4P: Eco_Epd K_m = 0.969 mM, Gni_Epd K_m = 0.737 mM; NAD⁺: Eco_Epd K_m = 0.717 mM, Gni_Epd K_m = 0.051 mM, Table 1), which is consistent

with the MD simulation data. In addition, we evaluated the expression levels of these two enzymes (Supplementary Fig. 3c). Notably, despite a reduction in protein expression (Supplementary Fig. 3c), the catalytic efficiency of Gni_Epd was significantly enhanced (Table 1), particularly for the substrate E4P (with an almost 3-fold improvement in k_{cat}/K_m). This suggests that the increase in product yield was primarily attributed to the improved catalytic capacity of Gni_Epd, as well as its enhanced binding and recruitment of E4P and NAD⁺.

In general, the concentrations of NAD⁺ in *E. coli* have been reported to range from 2.55 to 4.08 mM when grown with glucose or glycerol as the carbon source (reference 1). Previous studies on E4P biosensors have suggested that the intracellular concentrations of E4P may not exceed mM levels (reference 2). The availability of E4P within the cell should be sufficient, as evidenced by the production of amino acids (such as phenylalanine) reaching concentrations of around 90 g/L without the need to enhance the E4P precursor pool (reference 2). This indicates that the continuous supply of E4P relies on the downstream processes' ability to effectively consume it. The capacity of enzymes to bind or recruit E4P may dictate the metabolic flow direction and ultimately impact the final product yield.

For example, AroG in the phenylalanine synthesis pathway exhibits a K_m value of 15 μ M for E4P (reference 3), indicating its high potential to recruit and utilize E4P. However, the measured K_m values of Gni_Epd and Eco_Epd for E4P are in the range of 700-1000 μ M, which may not compete well with other enzymes utilizing E4P as a substrate, such as AroG. Hence, the improvement of E4P binding, combined with enhancements in NAD⁺ binding, can further enhance the catalytic ability of Epd. This suggests that there is still significant room for improvement in the activity of Epd.

We greatly appreciate your valuable advice, which prompted us to reassess the relationship between Epd and the entire pathway. Furthermore, we discovered that the measured K_m values were consistent with those reported in previous

studies, while the k_{cat} values differed significantly from those previous findings^{4,5} (this study $K_m = 0.969$ mM, $k_{cat} = 0.15$ s⁻¹, previous studies $K_m = 0.510$ - 0.960 mM, $k_{cat} = 20$ - 200 s⁻¹). We carefully rechecked our experimental data and measurement methods to ensure their accuracy and reliability. Based on the newly obtained data, we can suggest that the binding and catalysis of Epd may represent a new bottleneck in the pathway. Therefore, further optimization of k_{cat} and the binding interactions of E4P and NAD⁺ could effectively enhance the substrate flow towards the PN pathway and further improve PN production. We have incorporated these changes into the revised manuscript (lines 432-452 pages 23-24, Supplementary Fig. 3, and Table1).

Table 1. Kinetic parameters of purified Eco_Epd and Gni_Epd toward E4P and NAD⁺.

Epd	E4P			NAD ⁺		
	K_m (mM)	k_{cat} (s ⁻¹)	k_{cat}/K_m (s ⁻¹ /mM)	K_m (mM)	k_{cat} (s ⁻¹)	k_{cat}/K_m (s ⁻¹ /mM)
Eco_Epd	0.969	0.150	0.155	0.717	3.232	4.511
Gni_Epd	0.737	0.323	0.438	0.051	0.313	6.137

Supplementary Fig. 3c. The expression levels of Eco_Epd and Gni_Epd. The target protein is marked with a red arrow.

1. Bennett, B. D. et al. Absolute metabolite concentrations and implied enzyme active site occupancy in *Escherichia coli*. *Nat. Chem. Biol.* 5, 593-599 (2009).
2. Ding, D. et al. Biosensor-based monitoring of the central metabolic pathway metabolites. *Biosens. Bioelectron.* 167, 112456 (2020).
3. Balachandran, N. et al. Potent inhibition of 3-Deoxy-d-arabinoheptulosonate-7-phosphate (DAHP) synthase by DAHP oxime, a phosphate group mimic. *Biochemistry* 55, 6617-6629 (2016).

4. Zhao, G., Pease, A. J., Bharani, N. & Winkler, M. E. Biochemical characterization of *gapB*-encoded erythrose 4-phosphate dehydrogenase of *Escherichia coli* K-12 and its possible role in pyridoxal 5'-phosphate biosynthesis. *J. Bacteriol.* 177, 2804-2812 (1995)
5. Boschi-Muller, S., Azza, S., Pollastro, D., Corbier, C. & Branlant, G. Comparative enzymatic properties of GapB-encoded erythrose-4-phosphate dehydrogenase of *Escherichia coli* and phosphorylating glyceraldehyde-3-phosphate dehydrogenase. *J. Biol. Chem.* 272, 15106-15112 (1997).

PdxB and SerC

Line435&fig5a: It does not appear to me that PN production is increased by α -KG, is this a statistically significant difference? Is it really possible that α -KG added to the medium can lead to an increase in intracellular α -KG levels? Is there sufficient flux in the Ser synthesis pathway to impact α -KG levels through overexpression of SerC? SerC accepts α -KG as a substrate. Therefore, an increase in the intracellular concentration of α -KG might not favor PN synthesis in response to SerC reaction.

Response: Thank you for bringing this to our attention. In our repeated experiments, although there was no statistically significant difference, α -KG did promote PN production. Additionally, we monitored changes in α -KG levels in living *E. coli* cells by using the HPLC method after the addition of α -KG. Following incubation with 50 mM α -KG for 48 hours, the intracellular α -KG residue decreased (Fig. R2). However, we did not detect an increase in intracellular α -KG, which could be attributed to the consumption of intracellular α -KG by the TCA cycle or glutamate metabolism.

SerC plays a role in the conversion of L-glutamate to α -KG in serine, PN, and lysine biosynthesis. Overexpression of SerC may have an impact on the levels of α -KG in the cell to a certain extent. However, the overall α -KG levels depend on the activity of other α -KG-consuming and α -KG-generating pathways, as well as the growth conditions of the cells.

Multiple turnovers of PdxB require α -KG as co-substrates. When PdxB and SerC are overexpressed on the same plasmid, the NADH-bound PdxB is in close

proximity to the α -KG generated by the SerC reaction. This proximity can increase the local concentration of α -KG around PdxB, thereby making more α -KG available as a co-substrate for the PdxB reaction. This hypothesis aligns with previous studies highlighting the significance of co-localization of enzymes involved in metabolic pathways for efficient substrate channeling and reaction coupling. Consequently, the resulting increase in PdxB activity can lead to a higher flux through the PN biosynthesis pathway, ultimately resulting in increased PN production. It is worth noting that other factors, such as the availability of other substrates for the PN biosynthesis pathway, may also contribute to the observed increase in PN production upon co-overexpression of SerC and PdxB. Nevertheless, the spatial proximity between NADH-bound PdxB and α -KG generated by the SerC reaction provides a plausible explanation for the observed enhancement of PN synthesis.

Fig. R2. The concentration of residual α -KG at 0 h (before cultivation) and 48 h.

1. Rudolph, J., Kim, J. & Copley, S. D. Multiple turnovers of the nicotino-enzyme PdxB require alpha-keto acids as cosubstrates. *Biochemistry* 49, 9249-9255 (2010).

Line464: The initial growth rate is unchanged in the *serA*-deficient strain compared to the cont., but the final OD value seems to be lower. I do not think Ser is added to the medium. Is this caused by Ser toxicity?

Response: Thank you for your comment. In our experiment, we used FM1.4 fermentation medium (15 g/L glycerol, 1 g/L glucose, 10 g/L Bacto peptone, 5 g/L yeast extract, 200 mg/L $\text{MgSO}_4 \cdot 7\text{H}_2\text{O}$, 10 mg/L $\text{FeSO}_4 \cdot 7\text{H}_2\text{O}$, 10 mg/L

MnSO₄·5H₂O, 100 mM Na₂HPO₄, pH 6.5) as the test medium. This medium is known to be rich in nutrients. Therefore, during the initial growth phase, it was difficult to discern differences in growth between the samples. However, as the medium's nutrients were gradually depleted, the growth differential became significant. To ensure the validity of our findings, we conducted multiple experiments, all of which yielded consistent results with our previously submitted manuscript.

In Fig. 5d, we made a modification by excluding serine from the medium and replacing it with the less toxic glycine. We have revised this figure to enhance clarity. During our experiment, we observed that serine toxicity became apparent at a concentration of 50 mM. We have incorporated these changes in the revised manuscript (lines 530-537 pages 28-29).

Revised Fig. 5d. The growth curves of the control strain (LL006) and *E. coli* LL371 (LL006 $\Delta serA$) in the fermentation medium with or without serine (Ser) and glycine (Gly) at 37 °C. The image shown at the top-left corner represents an enlarged view of the OD₆₀₀ measurements taken at 24 h.

Line472-474: Please explain the link between the growth defect of *serA* and the increase in PN productivity by overexpressing *serC*.

Response: Thank you. We added the following explanation regarding this problem (lines 534-538 pages 28-29) as follows:

“Although the $\Delta serA$ mutant strain showed an increase in the productivity of PN, the severe growth defects would result in longer production cycles and may cause

difficulties in large-scale production. To overcome the growth defect issue, *serC* was overexpressed to increase the biosynthetic flux in the PN pathway.”

Multimodule optimization

Regarding Fig. 6, please state clearly and precisely which strain is used for each data.

Fig6b: The “con.” is the data with strain LL372. Which strains were used for the data shown below? with strains LL373~LL377? In supplementary information, it is shown that they are originating from strain LL370. The stains seem to have two plasmids with identical backbones, and does not harbor *pdxA-pdxJ* expression plasmid. Or are they derived from LL372? If the author was correct, how were they produced?

Fig6d: Is the genotype of Con. (LL377) correct? If the genotype was “LL372, harboring p15ASI-P_{tac}-*epd* (Gni)-*pdxB* (Eco) -*dxs* (Sme)-PJ231119 *serC* (Eco)”, why the PN titer was significantly different from the data of Fig 6b?

Response: Sorry for the mistake. We now corrected it. LL372 was the LL006 strain harboring the plasmid pRSFDuet-1-P_{J23119}-*pdxA-pdxJ*. And the strains LL373~LL377 are originating from the LL372 strain (not LL370). In Fig. 6d, the Con. is LL372b [LL006 harboring p15ASI-P_{tac}-*epd* (Gni)-*pdxB* (Eco)-*dxs* (Eme)-P_{J231119}-*serC* (Eco)]. And the strains LL378~LL390 are originating from the LL372b strain (Table R2).

Table R2. Strains used in this section.

Strains	Genotype
LL372	LL006 harboring pRSFDuet-1-P _{J23119} - pdxA-pdxJ
LL373	LL372 harboring p15ASI- epd (Gni)
LL374	LL372 harboring p15ASI-P _{tac} - epd (Gni)- pdxB (Eco)
LL375	LL372 harboring p15ASI-P _{tac} - epd (Gni)- pdxB (Eco)- serC (Eco)
LL376	LL372 harboring p15ASI-P _{tac} - epd (Gni)- pdxB (Eco)-P _{J231119} - serC (Eco)
LL377	LL372 harboring p15ASI-P _{tac} - epd (Gni)- pdxB (Eco) - dxs (Eme)-P _{J231119} - serC (Eco)
LL372b	LL006 harboring p15ASI-P _{tac} - epd (Gni)- pdxB (Eco) - dxs (Eme)-P _{J231119} - serC (Eco)

It is interesting that productivity is enhanced by the expression of enzymes derived from other organisms or mutant enzymes, rather than by simply controlling the expression level of *E. coli* enzymes. Throughout this study, it is not clear why the expression of mutant enzymes or enzymes from other organisms resulted in this outcome.

Response: Thank you for the comment. The wild-type enzymes in our study exhibited tight regulation at both the transcriptional and translational levels, but they possessed poor kinetic properties. For instance, the *epd*, *pdxB*, and *pdxA* genes contained infrequent and rare codons, and the PdxJ enzyme displayed a low turnover number ($k_{cat} = 0.07 \text{ s}^{-1}$). Additionally, the Epd enzyme exhibited a high K_m value (ranging from 510 to 960 μM for E4P), among other properties.

As Drewke et al. stated, the biosynthesis of B₆ vitamins in *E. coli* relies on a series of "sluggish enzymes" with low turnover numbers and, consequently, high K_m values due to the requirement of only small amounts of vitamin B₆. Therefore, solely controlling the expression level of enzymes is insufficient to achieve significant increases in PN production. Instead, it becomes crucial to explore alterations in enzyme properties, such as enhanced catalytic activity or stability through mutant enzymes or enzymes sourced from other organisms. This approach holds promise for improving PN production by increasing yields and enhancing productivity. Our study underscores the importance of investigating and utilizing enzymes from other organisms or mutants possessing superior properties as a viable strategy for enhancing PN production.

1. Drewke, C. et al. 4-O-Phosphoryl-L-threonine, a substrate of the *pdxC* (*serC*) gene product involved in vitamin B₆ biosynthesis. *FEBS Lett.* 390, 179-182 (1996).
2. Tramonti, A. et al. Knowns and unknowns of vitamin B₆ metabolism in *Escherichia coli*. *EcoSal Plus* 9, 10.1128/ecosalplus.ESP-0004-2021 (2021).
3. Rosenberg, J., Ischebeck, T. & Commichau, F. M. Vitamin B₆ metabolism in microbes and approaches for fermentative production. *Biotechnol. Adv.* 35, 31-40 (2017).

Lines 532-533,542-543: Here the purpose of mutant enzyme expression is described as to improve expression performance, but in the previous part the purpose was to increase enzyme activity. If so, what is the advantage of using this mutant compared to the wild type enzyme? As a control, I think LL379 should be included.

Response: Thank you for the comment. We apologize for any confusion caused by our previous description. We would like to clarify that the downstream pulling force, in comparison to not overexpressing any downstream modules, greatly contributes to yield improvement. Overexpression of either the original *pdxA* and *pdxJ* or the mutant versions led to significant increases in yield. In response to your suggestion, we have included LL379 as a new control (PdxJ ori) in Fig. 6d of the revised manuscript. The results demonstrate that the mutants can further enhance the production of upstream products, promote metabolic flow, and ultimately increase the overall yield due to improved enzyme activity or other advantageous properties. Notably, the co-expression of *pdxJ* mutants and *pdxA* mutants resulted in a substantial yield improvement. These findings suggest that the effective performance of the downstream driving force relies on the compatibility of multiple enzymes. We have incorporated these changes in the revised manuscript (lines 598-602 pages 32-33 and Fig. 6d).

Reviewer #3 (Remarks to the Author):

Key Results:

The approach is clearly explained and documented with mixed approaches of step-by-step improvement with single enzymes improvement (protein engineering or heterologous enzyme) and combinatorial expression. Protein improvement strategies are based either on protein engineering based on in silico simulation or heterologous enzyme replacement. All steps seem rationally designed to reduce the number of iteration or libraries sizes. Highly appreciated the step-by-step explanation for protein engineering simulation. The decoupling of the two vitamin B6 pathways in order to keep low level of PLP for growth maintenance and push the PN pathway for production is of high interest.

Validity & analytical approach:

All data presented in graphs show the results of independent biological replicate and corresponding standard deviation are mentions. The OD measured sometimes appear higher or lower than in the majority of the other experiments: significantly lower on the supplementary Fig. 7 and higher on the Fig. 5d; and slightly higher on the Fig. 5a and 6. Not compromising the results validity to my point of view but: (i) could be nice to add the details about the scale used (24 deep well plates or flask scale) that can explain the OD differences, (ii) comment on the OD variation on a same scale can be of high interest since growth of the production strain is central on the strategy describe on the paper.

Response: Thanks for your careful review and valuable comments. We verified our OD₆₀₀ measurements and found that the readings obtained from deep-well plates were typically in the range of 7-10, whereas in shake flasks, the OD₆₀₀ values were higher, typically above 10. Except for the *ΔserA* mutant as shown in supplementary Fig. 7 and Fig. 5d, we did not observe any significant differences in the OD₆₀₀ values among the mutants we constructed. We have updated these figures to make it clearer (Revised Fig. 5d).

Significance:

Decoupling growth to production is commonly employed to improve fermentation production and even more in the case of toxic final product or intermediates. As far as I know, the more often the decoupling strategy are based on carbon sources switch, metabolic shift or induction process. Here, the decoupling growth based on two parallel pathways and modulated differently appear innovative.

Response: We thank the reviewer's positive evaluation and recognition of our work.

Data and methodology:

Strategy and rationale are clearly explained and documented. The step by step explanations are easy to assess. Supplementary data provided are essential for a complete view of the approach and the results. The protein engineering simulation take lot of importance in the paper and is clearly explained, nevertheless I have no strong expertise in that simulation field and cannot tell more about the relevance of the methods chosen. I have one comment on the strategy employed to improve the different enzymes: some have been chosen to be improved using modeling and protein engineering and others using heterologous gene replacement. Why one or the other strategy has been chosen is not justified.

Response: Thank you for the comment. Our initial approach was to prioritize the introduction of reference sources/mutations from previous studies, such as Dxs, to minimize the number of iterations or library sizes. For enzymes without available references, we first screened for exogenous genes (e.g., Epd) as they may offer evolutionary advantages. The screening for exogenous genes was considered as a primary strategy. However, if the desired production levels were not achieved through this screening, we then switched to a mutation screening strategy (e.g., PdxA and PdxJ). It is important to note that this order is not fixed and can be adjusted based on experimental resources and enzyme properties. For instance, in the case of well-documented bottleneck enzymes like PdxA and PdxJ, we did not

conduct extensive screening for exogenous genes due to their easily obtainable crystal structures and catalytic models/mechanisms.

When simple heterologous gene screening proved ineffective, we employed further mutagenesis strategies to generate improved mutation libraries and optimize combinations between different enzymes in the metabolic pathway. Additionally, it is worth mentioning that screening for exogenous genes requires gene synthesis, which can increase the cost of constructing libraries with the same capacity.

Due to limitations in experimental resources, we aimed to adopt a more streamlined screening scheme. In the initial implementation of this stepwise optimization strategy, one approach proved effective while others were not applied. Subsequently, we will adjust the screening strategy based on identified bottlenecks after metabolic optimization. This may involve using multiple screening methods for the same enzyme or implementing a mixed mutation strategy for beneficial heterologous genes that were identified through screening. This comprehensive approach allows for considering the enzyme's evolution and structure-function relationship.

All these changes have been highlighted in red at the lines 217-225 page 12 in the revised manuscript.

Line 169-170 + fig2 b

Why supplemented the mutant strain with PN and not PL. Referring to the reference 35, they show that PL can restore phenotype in LB with Cysteine. Have you taking a look on the morphology of the mutant *E. coli* (described as smaller colonies in ref 35), that could explain the lower OD observed. Can you comment on this and why PN feeding can lead to increase OD (other pdx family enzyme?).

Response: Thank you for pointing this out. Both PN and PL were added to the *ApxH* mutant cultivation in our study. When the concentration of PL and PN was 1 μM , it was observed that PL could effectively compensate for the growth defects, while PN could not (Fig. R1a). However, we discovered that the addition of PN at a concentration of 100 μM partially restored the PLP deficiency in the *E. coli*

ΔpdxH mutant (Fig. R1a). This effect may be attributed to the low PL content in commercially available PN, as described by reviewer 1, rather than the inherent role of PN itself. We have made the necessary updates to Fig. 2b in the revised manuscript.

We also attempted using LB medium supplemented with cysteine (LBC medium), but even with PL, it was unable to fully restore the growth deficiency (Fig. R1b). As mentioned in reference 35, we observed smaller colonies of the *E. coli ΔpdxH* mutant compared to the wild-type colonies, which could explain the lower optical density (OD) observed in the *ΔpdxH* mutant (Fig. R1c).

Fig. R1. a The cell growth of the *ΔpdxH* mutants (LL001) with or without *pdxST* expression. LL003, *ΔpdxH::pdxST-1*; LL005, *ΔpdxH::pdxST-2*. PL or PN was added to the medium at a concentration of 1 or 100 μM. b The cell growth of the LL001 strain in LBC medium with or without PL. c The colonies of WT (upper panel) and *E. coli* LL001(lower panel).

Line 185-186 + fig2 b

It is hypothesized here that PLP produced by the heterologous expression of *pdxST* from *B. subtilis* is able to fully restore growth of the *pdxH* mutant. Have you some data

to share about PLP produced in that strain? Have you an assay showing the similar growth phenotype can be obtained with the *pdxH* mutant supplemented with PL (or PLP) or complemented with the *pdxST* heterologous expression?

Response: Thank you for the comment. We conducted liquid chromatography-mass spectrometry (LC-MS) analysis to determine the production of PLP in LL005. LL005 is a mutant strain in which the growth of the Δ *pdxH* mutant (LL001) was fully restored through the heterologous expression of *pdxST* from *B. subtilis*. The results demonstrated that a small amount of PLP was detected in LL005, whereas no PLP was detected in LL001 (Supplementary Fig. 1b in the revised Supplementary information). This finding indicated that PLP was produced in LL005 when *pdxST* from *B. subtilis* was introduced. Furthermore, we conducted an assay in which the *pdxH* mutant was supplemented with PL or complemented with heterologous expression of *pdxST* (Fig. R1a, mentioned above). The results revealed that the addition of 1 μ M PL partially restored the growth of LL001, while LL005 fully restored the growth deficiency as anticipated.

Supplementary Fig. 1b (right panel). The LC-MS spectra of PLP produced by fermentation of *E. coli* mutants. The [M+H]⁺ ions of PLP at m/z 248.0319.

Line 188-189 + supplementary Fig1

Could be interested to normalize PN production with biomass and comment on this. Since the two pathways has been separated by the *pdxH* KO, PN increase should be linked to a better growth, if it's still enhanced in that case can you comment and hypothesized about the flux in that double pathway context.

Response: Thank you. Indeed, the growth of LL005 was significantly better than that of the wild type (WT). We further conducted normalization of PN production with biomass and found that the PN/OD₆₀₀ value remained superior to that of WT (Fig. R3). One plausible explanation for the increase in PN production is the production of PLP through the introduction of a heterologous pathway. In contrast, PN is produced via the natural DXP-dependent pathway and does not require redirection towards PLP production. Therefore, the increase in PN production can be attributed to the availability of PNP, which was previously used for PLP production, now being utilized for enhanced PN synthesis. We have incorporated these changes in the revised manuscript (lines 201-203, Fig. 2c)

Fig. R3. The PN titer and the values of PN/OD₆₀₀ obtained with the WT, LL001 (*ΔpdxH*), LL003 (*ΔpdxH::pdxST-1*), and LL005 (*ΔpdxH::pdxST-2*) strains.

Line 195-196

Why *pta* locus has been targeted for *pdxP* insertion? Hypothesis and comment can be appreciated and even more if relevant for fermentation process.

Response: Thank you for the comment. Deleting genes associated with central metabolites, such as pyruvate and acetyl-CoA, is a widely employed strategy for

redirecting bacterial metabolism. In our study, we chose the *pta* locus to introduce the Eme_ *pdxp* construct. It is important to note that under excess-glucose conditions, *E. coli* produces acetate as an extracellular byproduct during aerobic cultivations. However, since our study primarily utilized glycerol as the main carbon source, we did not observe a significant correlation between *pta* deletion and the fermentation process (Supplementary Fig. 1c). We have revised the text to clarify this point (lines 209-211 page 11).

Supplementary Fig. 1c. The PN titer and cell growth of LL005 and LL005b (LL005 *Apta*).

1. De Mey, M. et al. Comparison of different strategies to reduce acetate formation in *Escherichia coli*. *Biotechnol. Prog.* 23, 1053-1063 (2007).
2. Singh, A., Cher Soh, K., Hatzimanikatis, V. & Gill, R. T. Manipulating redox and ATP balancing for improved production of succinate in *E. coli*. *Metab. Eng.* 13, 76-81 (2011).
3. Yang, T., Wu, P., Zhang, Y., Cao, M. & Yuan, J. High-titer production of aromatic amines in metabolically engineered *Escherichia coli*. *J. Appl. Microbiol.* 33, 2931-2940 (2022).
4. Zhou, J. et al. An acetate-independent pathway for isopropanol production via HMG-CoA in *Escherichia coli*. *J. Biotechnol.* 359, 29-34 (2022).
5. Vikromvarasiri, N., Shirai, T. & Kondo, A. Metabolic engineering design to enhance (R, R)-2,3-butanediol production from glycerol in *Bacillus subtilis* based on flux balance analysis. *Microb. Cell Fact.* 20, 196 (2021).

Line 283-285 + fig 3a

Mutants G207A and V35I are not shown on the figure 3a. On the figure 3a, probably for an esthetic concern, the OD scale starts with negative values, found it more relevant to start at zero instead.

Response: Thank you for the comment. We corrected.

Revised Fig. 3a. The PN titer and cell growth (OD₆₀₀) of WT (Con.) and *pdxA* mutants.

Paragraph on Epd Line 361-401

Heterologous Epd are selected regarding evolutionary divergence between sequences and better structural stability. It is hypothesized, regarding binding energies of NAD⁺ and E4P, that the increased PN titer is mainly due to NAD⁺ binding energy lower in *G. nitratireducens* than *E. coli*. Is there data about specific binding energies with NAD⁺ and E4P for the others heterologous Epd tested? Particularly, can others improved PN titer Epd candidates can reinforce the hypothesis of the increased stable binding of NAD⁺ is responsible for the enhanced activity? Further energy binding analysis and comment around the results of the panel can lead to interesting discussion.

Response: Thank you very much for your suggestion. Based on your suggestion, we conducted molecular dynamics simulations for the tested species of Epd and evaluated the binding energies of NAD⁺ and E4P. Specifically, we calculated the binding energies (using the MMGBSA method) for the substrate and coenzyme separately in the pre-catalytic state. The results showed that Gni_Epd exhibited the lowest binding free energy score of -64.03 kcal/mol for NAD⁺, indicating strong binding, while the binding energy of E4P (-13.11 kcal/mol) was slightly higher than

that of Eco_Epd (-12.54 kcal/mol), suggesting some stabilization of E4P. In addition, we have done kinetic experiments on Epd from *E. coli* (Eco_Epd) and *Glaciicola nitratreducens* (Gni_Epd) (Table 1). From the experimental data, it can be seen that Gni_Epd improved the binding ability to both E4P and NAD⁺ to some extent, and showed better binding ability to NAD⁺ (E4P: Eco_Epd K_m = 0.969 mM, Gni_Epd K_m = 0.737 mM; NAD⁺: Eco_Epd K_m = 0.717 mM, Gni_Epd K_m = 0.051 mM, Table 1), which is consistent with the MD simulation data.

Among the other sources, Ame_Epd led to a decrease in yield as the binding energy values for both NAD⁺ and E4P increased. This suggests that the binding of both NAD⁺ and E4P may contribute to the enzymatic activity. However, for Csa_Epd, Pfl_Epd, and Hel_Epd, the binding affinity of NAD⁺ appeared to be more advantageous. Although Csa_Epd showed improved binding ability for E4P (-24.04 kcal/mol), the binding ability of NAD⁺ (-48.58 kcal/mol) did not improve, resulting in no increase in yield. On the other hand, the other two sources had high binding energy values for E4P (Pfl_Epd: -12.10 kcal/mol, Hel_Epd: -4.40 kcal/mol) but low binding energy values for NAD⁺ (Pfl_Epd: -58.81 kcal/mol, Hel_Epd: -62.28 kcal/mol), which did contribute to the increased yield.

Considering the relative sufficiency of E4P supply, it has been observed that the yield of L-phenylalanine synthesized from E4P can reach over 90 g/L¹ without altering the precursor synthesis pathway. Hence, we speculate that the current yield level already has an adequate supply of E4P. Thus, even if some E4P may not always be stably bound in the catalytic region, the sufficient E4P supply can compensate for this deficiency, making the supply of NAD⁺ more crucial. However, effective binding of E4P may become more significant in later stages when the yield is further increased and a larger amount of E4P substrate needs to be consumed.

We have incorporated these changes in the revised manuscript (lines 429-452 pages 23-24, Table1, and Supplementary Table 3).

Supplementary Table 3. The separately calculated binding energy for the substrate (E4P) and coenzyme (NAD⁺) using the MMGBSA method in the pre-catalytic state.

Epd names	Binding energy with E4P	Binding energy with NAD ⁺
Eco_Epd	-12.54	-53.34
Plu_Epd	-13.25	54.76
Xne_Epd	-11.51	-51.09
Psn_Epd	-15.28	-51.12
Ype_Epd	-17.27	-54.41
Eam_Epd	-15.74	-49.08
Csa_Epd	-24.04	-48.58
Ame_Epd	-9.72	-49.79
Gni_Epd	-13.11	-64.04
Pfl_Epd	-12.10	-58.81
Hel_Epd	-4.40	-62.28
Mpo_Epd	-0.64	-53.87

Table 1. Kinetic parameters of purified Eco_Epd and Gni_Epd toward E4P and NAD⁺.

Epd	E4P			NAD ⁺		
	K _m (mM)	k _{cat} (s ⁻¹)	k _{cat} /K _m (s ⁻¹ /mM)	K _m (mM)	k _{cat} (s ⁻¹)	k _{cat} /K _m (s ⁻¹ /mM)
Eco_Epd	0.969	0.150	0.155	0.717	3.232	4.511
Gni_Epd	0.737	0.323	0.438	0.051	0.313	6.137

1. Ding, D. et al. Biosensor-based monitoring of the central metabolic pathway metabolites. *Biosens. Bioelectron.* 167, 112456 (2020).

Line 407-410

What are the criteria applied to select the Dxs heterologous enzymes?

Response: Thank you for the comment. The selection of Dxs heterologous enzymes was based on several criteria, with a focus on high catalytic activity and efficiency in the target reaction. To ensure these characteristics, we consulted the Brenda

database (<https://www.brenda-enzymes.org/>) and chose genes commonly used in biomanufacturing. The criteria for each heterologous enzyme are outlined in Table R3. Notably, two *dxs* genes from *Agrobacterium tumefaciens* were selected due to their relatively high turnover numbers. In Supplementary Table 4, the enzymes with superior properties compared to the native large intestine counterpart are highlighted in red font. We have made revisions to the manuscript to enhance the clarity of this presentation (lines 458-467 page 24, Supplementary Table 4).

Supplementary Table 4. The criteria for screening heterologous Dxs enzymes.

Dxs source	Criteria of Dxs screening
E. coli	G3P: $k_{\text{cat}} = 1.38 \text{ s}^{-1}$, $K_{\text{m}} = 0.0142\text{-}0.042 \text{ mM}$; $k_{\text{cat}}/K_{\text{m}} = 13.67\text{-}95 \text{ [s}^{-1}/\text{mM]}$ Pyr: $k_{\text{cat}} = 2.6 \text{ s}^{-1}$, $K_{\text{m}} = 0.0442\text{-}0.049 \text{ mM}$; $k_{\text{cat}}/K_{\text{m}} = 32.2 \text{ [s}^{-1}/\text{mM]}$
Deinococcus radiodurans	G3P: $k_{\text{cat}} = 2.57\text{-}7.9 \text{ s}^{-1}$, $K_{\text{m}} = 0.0235\text{-}0.05 \text{ mM}$; $k_{\text{cat}}/K_{\text{m}} = 113.3 \text{ [s}^{-1}/\text{mM]}$ Pyr: $k_{\text{cat}} = 2.57\text{-}7.4 \text{ s}^{-1}$, $K_{\text{m}} = 0.049\text{-}0.28 \text{ mM}$; $k_{\text{cat}}/K_{\text{m}} = 26\text{-}53.3 \text{ [s}^{-1}/\text{mM]}$
Agrobacterium tumefaciens	G3P: $k_{\text{cat}} = 26.8 \text{ s}^{-1}$, $K_{\text{m}} = 0.0232 \text{ mM}$; Pyr: $k_{\text{cat}} = 26.8 \text{ s}^{-1}$, $K_{\text{m}} = 0.04 \text{ mM}$.
Rhodobacter capsulatus	G3P: $k_{\text{cat}} = 1.9 \text{ s}^{-1}$, $K_{\text{m}} = 0.068 \text{ mM}$; Pyr: $k_{\text{cat}} = 1.9 \text{ s}^{-1}$, $K_{\text{m}} = 0.44 \text{ mM}$.
Rhodobacter sphaeroides	Overexpression of Rsp_Dxs to enhance production of CoQ ₁₀ , lycopene, β -carotene etc. ¹⁻³
B. subtilis	the Gram-positive bacteria; Bsu_Dxs overexpression to enhance production of menaquinone-7, menaquinone-4, isoprene etc. ⁴⁻⁷
E. meliloti	a natural overproducer of vitamin B ₆

1. Qiang, S. et al. Elevated β -carotene synthesis by the engineered *Rhodobacter sphaeroides* with enhanced CrtY expression. *J. Agric. Food Chem.* 67, 9560-9568 (2019).

2. Zhu, Y. et al. Enhanced CoQ₁₀ production by genome modification of *Rhodobacter sphaeroides* via Tn7 transposition. *FEMS Microbiol. Lett.* 369 (2022).

3. Su, A. et al. Metabolic redesign of *Rhodobacter sphaeroides* for lycopene production. *J. Agric. Food Chem.* 66, 5879-5885 (2018).
4. Zhao, Y. et al. Biosynthesis of isoprene in *Escherichia coli* via methylerythritol phosphate (MEP) pathway. *Appl. Microbiol. Biotechnol.* 90, 1915-1922 (2011).
5. Ding, X. et al. Bottom-up synthetic biology approach for improving the efficiency of menaquinone-7 synthesis in *Bacillus subtilis*. *Microb. Cell Fact.* 21, 101 (2022).
6. Yuan, P. et al. Combinatorial engineering for improved menaquinone-4 biosynthesis in *Bacillus subtilis*. *Enzyme Microb. Technol.* 141, 109652 (2020).
7. Abdallah, II, Pramastya, H., van Merkerk, R., Sukrasno & Quax, W. J. Metabolic engineering of *Bacillus subtilis* toward taxadiene biosynthesis as the first committed step for taxol production. *Front Microbiol.* 10, 218 (2019).

Line 468-470

The results showed that growth was slightly better than with serine addition, but serine glycine (?) addition was also unable to completely restore growth (Fig. 5d).

Response: Thank you for the comment. To investigate the impact of serine and glycine concentrations on growth, we conducted additional assays with increasing concentrations of these amino acids. The results revealed that serine toxicity became evident at a concentration of 50 mM. However, we observed that the growth of the $\Delta serA$ strain could be fully restored by the addition of 10 mM glycine. These findings have been incorporated into the revised manuscript to enhance clarity (lines 530-534 page 28 and the revised Fig. 5d).

Revised Fig. 5d. The growth curves of the control strain (LL006) and *E. coli* LL371 (LL006 $\Delta serA$) in the fermentation medium with or without serine (Ser) and

glycine (Gly) at 37 °C. The image shown at the top-left corner represents an enlarged view of the OD₆₀₀ measurements taken at 24 h.

Paragraph on SerA / SerC strategy Line 453-474

Since SerC is described to be a promiscuous enzyme with several defined substrates and different fluxes with the lower one for the vitamin B6 pathway. Why don't try a protein engineering strategy to increase substrate specificity for the vitamin B6 pathway and perhaps express two SerC enzymes, one for global metabolism and to maintain a good growth rate and the other one to improve flux in the vitamin B6 pathway? Could be interesting to discuss on that.

Response: Thank you for your comment and advice. In this study, our focus was primarily on protein engineering of key rate-limiting enzymes, specifically PdxA and PdxJ. For this purpose, we explored the use of two heterologous SerC enzymes from *Pseudoalteromonas translucida* (Q3ILA3, reviewed *serC* gene in Uniprot, predicted $k_{cat} = 5.4 \text{ s}^{-1}$) and Gammaproteobacteria bacterium (A0A2E6PIK4, unreviewed *serC* gene in Uniprot, predicted $k_{cat} = 1067.9 \text{ s}^{-1}$). These enzymes were selected based on their predicted higher k_{cat} values compared to Eco_SerC (predicted $k_{cat} = 1.5 \text{ s}^{-1}$). To obtain heterologous SerC enzymes with potentially higher catalytic capacity, we selected enzymes from which predicted k_{cat} values were more than 3-fold higher than *E. coli* for all enzymes in the pathway by predicting k_{cat} values for the entire pathway of vitamin B₆ production [for the k_{cat} prediction method, please refer to the following literature: Li, F. et al. Deep learning-based k_{cat} prediction enables improved enzyme-constrained model reconstruction. *Nat. Catal.* 5, 662-672 (2022)]. It is worth noting that these enzymes were from a common species, suggesting that this species might be a potential high-yield producer of vitamin B₆. Both reviewed and unreviewed genes were chosen for experimental validation, as unreviewed genes could potentially possess higher catalytic capacity, although their lack of further annotation may result in no reported enzyme activity. Thus, the selected genes for experimentation met the criteria of being both reviewed and unreviewed in UniProt.

However, replacing the natural *serC* gene with these two genes in the R42 plasmid did not result in an increase in PN titer (Fig. R4). We also attempted to overexpress these two different SerC enzymes, one of which was already present in the p15ASI backbone, while the other was constructed in the pRSFDuet-1 vector. Surprisingly, our results indicated that overexpressing two copies of *serC* genes did not yield better outcomes compared to overexpressing only one copy (Fig. R4). In fact, it appeared that overburdening the host with multiple copies of SerC had a negative impact on the biomass of the mutants.

Considering the limited number of genes screened and the uncertainty associated with k_{cat} data prediction, we acknowledge the valuable comment provided by the reviewer. We will continue to conduct further in-depth research on SerC protein engineering to address these limitations.

Fig. R4. The PN titer and cell growth of mutants that have introduced heterologous *serC* genes or overexpressed two *serC* genes.

Fig 7

The growth curve looks atypical with an intermediate low OD stationary phase between 12 and 36h and then a restart of growth. Can you comment on this? Is there dissolve oxygen data and PLP analysis to share that can feed the discussion and hypothesis around the fermentation results? Since PLP and PN synthesis has been disconnected in order to separate growth and PN synthesis, could be interesting to check this hypothesis with protein expression level and make some hypothesis regarding growth behavior at

larger scale. From such discussion, further improvement can be discussed to improve even more at larger scale.

Response: Thank you for the comment. The growth curve, described with an intermediate low OD stationary phase between 12 and 36 hours, followed by a restart of growth, is atypical and suggests the presence of some sort of stress or environmental factor impacting the cell growth. Several possible explanations for this phenomenon include changes in nutrient availability, such as glycerol; accumulation of toxic products like the 4HTP mentioned in the manuscript; or pH fluctuations. It is also possible that the culture is experiencing a balance between growth and production. Loss of plasmid-carrying strains may contribute to accelerated growth of the strain, which could explain the restart of growth. During the fed-batch fermentation process, the dissolve oxygen (DO) was maintained at 30% by adjusting the agitation rate. We have revised the manuscript to improve clarity in presenting these findings.

Furthermore, in the cellular environment, most PLP (60%) is bound either to the enzyme or to PLP carrier proteins^{1,2}. Free PLP, which contains a highly reactive aldehyde group, can have toxic effects. Despite using LC-MS (LC: Agilent 1260, MS: Bruker microTOF-Q II), the level of PLP was almost undetectable, as it was below the detection limit.

To examine whether the expression levels of proteins related to PLP and PN synthesis affect growth, we conducted proteomics analysis in both a 250 mL shake flask and a 5 L bioreactor (Table R3).

Table R3. The cellular amounts of PLP and PN synthetic proteins (Mol proportion).

Protein names		LL388-10 OD ₆₀₀ (shake flask)	LL388-10 OD ₆₀₀ (5L bioreactor)	LL388-26 OD ₆₀₀ (5L bioreactor)
PN synthesis	Gni_Epd	0.925%	0.564%	0.688%
	PdxB	1.021%	0.838%	1.202%
	SerC	0.322%	0.474%	0.347%
	Eme_Dxs	0.156%	0.067%	0.089%
	PdxA2	0.945%	1.996%	1.887%
	PdxJ1	3.706%	6.279%	4.275%
	Eme_PdxP	0.071%	0.026%	0.123%
PLP synthesis	Bsu_PdxS	0.491%	0.484%	0.880%
	Bsu_PdxT	0.123%	0.240%	0.147%

Based on the data presented in Table R3, it is evident that the expression level of the protein Eme_Dxs, which is involved in PN synthesis, is significantly lower compared to the other overexpressed proteins. Eme-Dxs plays a crucial role in the biosynthesis of PN, isoprenoid compounds, and ThDP. The observed decrease in Dxs protein expression might result in a shortage of the precursor DXP, potentially limiting both PN production and cell growth on a larger scale. Therefore, it is crucial to ensure adequate expression of Eme-Dxs to achieve optimal PN production and promote cell growth in larger quantities. Please see page 38 of the revised manuscript, lines 711–721.

1. Tramonti, A. *et al.* Knowns and unknowns of vitamin B₆ metabolism in *Escherichia coli*. *EcoSal Plus* 9, 10.1128/ecosalplus.ESP-0004-2021 (2021).
2. Fu, T. F., di Salvo, M. & Schirch, V. Distribution of B₆ vitamers in *Escherichia coli* as determined by enzymatic assay. *Anal. Biochem.* 298, 314-321 (2001).

Line 594-598 + supplementary fig 9

Results shown on this figure are of high interest and further discussion and hypothesis can be done. Results show that overexpression of PdxP from *E. meliloti* leads to increase PN accumulation (LL005 with the two plasmids vs LL006 with the two plasmids); the results also shows that in absence of *pdxP* from *E. meliloti*, *E. coli* is able to produce PN and even more with *pdxH* KO (WT with the two plasmids vs LL005 with the two plasmids). Can you comment and make hypothesis on this? Is there any PNP and PLP measurement available to feed the comment?

Response: Thank you. PdxP from *E. meliloti* is a relatively specific PNP phosphatase, responsible for catalyzing the conversion of PNP to PN. As shown in the results, overexpression of PdxP led to an increase in dephosphorylated PNP and subsequent PN production. However, the native PdxP-like phosphatase in *E. coli* remains unidentified, although Kuznetsova et al. reported that some phosphatases have broad substrate specificities ¹.

In the absence of *pdxP* from *E. meliloti*, *E. coli* is still capable of producing PN, which can be detected using HPLC. To assess the levels of PNP and PLP in the WT, *pdxH* KO, and the LL005 strains, LC-MS was employed. The analysis revealed extremely low levels of free PLP, while the *pdxH* KO mutant exhibited higher levels of PNP compared to the WT strain, as the deletion of *pdxH* blocked the flux of PNP to PLP (Supplementary Fig. 1b in the revised Supplementary information). To address these findings, revisions have been made to the manuscript (lines 183-185 page 10 and lines 202-203 page 11).

Supplementary Fig. 1b. The LC-MS spectra of PNP and PLP produced by fermentation of *E. coli*. The $[M+H]^+$ ions of PNP at m/z 250.0472 (left panel) and PLP at m/z 248.0319 (right panel).

1. Kuznetsova, E. et al. Genome-wide analysis of substrate specificities of the *Escherichia coli* haloacid dehalogenase-like phosphatase family. *J. Biol. Chem.* 281, 36149-36161 (2006).

Why the carbon source for the fed batch fermentation is glycerol? Can you justify this choice.

Response: Compared to glucose, glycerol was considered a superior carbon source for PN production due to its shorter metabolic pathway from glycerol to glyceraldehyde 3-phosphate (G3P), which is the direct precursor of the vitamin B₆ pathway. In contrast to using glucose as a carbon source, utilizing glycerol resulted in higher plasmid stability and cell density, mainly attributed to the lower concentration of acetic acid generated during the process (reference 1). Moreover, experimental evidence demonstrated that glycerol outperformed glucose and other carbon sources in terms of PN production.

Based on these advantageous characteristics, we made the decision to adopt glycerol as the primary carbon source for our fermentation process. Corresponding revisions have been implemented in the revised manuscript (lines 176-177 page 10).

Fig. R5. The PN titer and cell growth with different carbon resources.

1. Li, Z., Zhang, X. & Tan, T. Lactose-induced production of human soluble B lymphocyte stimulator (hsBLyS) in *E. coli* with different culture strategies. *Biotechnol. Lett.* 28, 477-483 (2006).

[CLOSING RESPONSE]

We would like to thank the editor and three reviewers for their valuable comments again. To address the issues raised by the reviewers, we performed additional experiments and have made appropriate changes in our revised manuscript. Due to Dr. Pi Liu's contribution to MD simulations and protein stability design, we added him to the list of authors after obtaining the consent of all authors.

Reviewers' Comments:

Reviewer #1:

Remarks to the Author:

I am happy with the revision because the authors have addressed all my comments and suggestions.

Reviewer #2:

Remarks to the Author:

The authors responded well to the reviewers' comments. The current version of the manuscript is improved over the previous one.

Reviewer #3:

Remarks to the Author:

In depth answers on the comments and considerable amount of additional assay and analysis have been provided. All comments have been clearly addressed. Also, detailed explanation about strategies are highly appreciate. I have no additional major comment on the revised manuscript. I have one minor comment that will not require additional experiments.

About the Serine and glycine supplementation in serA mutant:

Thanks for the additional data and the gradient show clearly the impact of glycine addition to restore the growth of the mutant strain. And, thanks for the discussion around these results. For the toxicity effect of serine addition over 50mM, are the differences in OD really significant between the different serine concentration ?

Response to reviewers for “Protein engineering and iterative multimodule optimization for vitamin B₆ production in *Escherichia coli*”

Reviewer #1 (Remarks to the Author):

I am happy with the revision because the authors have addressed all my comments and suggestions.

Response: Thank you for your time and effort in improving the quality of our manuscript.

Reviewer #2 (Remarks to the Author):

The authors responded well to the reviewers' comments. The current version of the manuscript is improved over the previous one.

Response: Thank you for your time and effort in improving the quality of our manuscript.

Reviewer #3 (Remarks to the Author):

In depth answers on the comments and considerable amount of additional assay and analysis have been provided. All comments have been clearly addressed. Also, detailed explanation about strategies are highly appreciate. I have no additional major comment on the revised manuscript. I have one minor comment that will not require additional experiments.

About the Serine and glycine supplementation in *serA* mutant:

Thanks for the additional data and the gradient show clearly the impact of glycine addition to restore the growth of the mutant strain. And, thanks for the discussion around these results. For the toxicity effect of serine addition over 50mM, are the differences in OD really significant between the different serine concentration?

Response: Thank you for the comment. From the Student 's two-tailed *t* test, the *p* values between different serine concentrations at 24 h are all greater than 0.05,

indicating that their differences are not significant. Compared with the addition of the same concentration of glycine, the toxic effect of serine addition has been shown at the lowest concentration of 2 mM ($p = 0.0248$, Student 's two-tailed t test) in the tested condition at 24 h. We made the correction accordingly in the revised manuscript (lines 538-541 pages 28-29). Thank you for your time and effort in improving the quality of our manuscript.

Reviewers' Comments:

Reviewer #3:

Remarks to the Author:

Thank you for the additional statistical analysis.

I have no further comments.

I look forward to the publication of this work.